# Schizophrenia diagnosis based on diverse epoch size resting-state EEG using machine learning



Athar Alazzawı[1], Saif Aljumaili[1], Adil Deniz Duru[2], Osman Nuri Uçan[1], Oğuz Bayat[1], Paulo Jorge Coelho[3,4] and Ivan Miguel Pires[5]

[1] Electrical and Computer Engineering, School of Engineering and Natural Sciences, Altınbaş University, Istanbul, Turkey
[2] Neuroscience and Psychology Research in Sports Lab, Faculty of Sport Science, Marmara University Istanbul, Istanbul, Turkey
[3] Polytechnic Institute of Leiria, Leiria, Portugal
[4] Institute for Systems Engineering and Computers at Coimbra (INESC Coimbra), Coimbra, Portugal
[5] Instituto de Telecomunicações, Escola Superior de Tecnologia e Gestão de Águeda, Universidade de Aveiro, Águeda, Portugal

Corresponding author
Ivan Miguel Pires, impires@it.ubi.pt

## ABSTRACT

Schizophrenia is a severe mental disorder that impairs a person's mental, social, and emotional faculties gradually. Detection in the early stages with an accurate diagnosis is crucial to remedying the patients. This study proposed a new method to classify schizophrenia disease in the rest state based on neurologic signals achieved from the brain by electroencephalography (EEG). The datasets used consisted of 28 subjects, 14 for each group, which are schizophrenia and healthy control. The data was collected from the scalps with 19 EEG channels using a 250 Hz frequency. Due to the brain signal variation, we have decomposed the EEG signals into five sub-bands using a band-pass filter, ensuring the best signal clarity and eliminating artifacts. This work was performed with several scenarios: First, traditional techniques were applied. Secondly, augmented data (additive white Gaussian noise and stretched signals) were utilized. Additionally, we assessed Minimum Redundancy Maximum Relevance (MRMR) as the features reduction method. All these data scenarios are applied with three different window sizes (epochs): 1, 2, and 5 s, utilizing six algorithms to extract features: Fast Fourier Transform (FFT), Approximate Entropy (ApEn), Log Energy entropy (LogEn), Shannon Entropy (ShnEn), and kurtosis. The L2-normalization method was applied to the derived features, positively affecting the results. In terms of classification, we applied four algorithms: K-nearest neighbor (KNN), support vector machine (SVM), quadratic discriminant analysis (QDA), and ensemble classifier (EC). From all the scenarios, our evaluation showed that SVM had remarkable results in all evaluation metrics with LogEn features utilizing a 1-s window size, impacting the diagnosis of Schizophrenia disease. This indicates that an accurate diagnosis of schizophrenia can be achieved through the right features and classification model selection. Finally, we contrasted our results to recently published works using the same and a different dataset, where our method showed a notable improvement.

## INTRODUCTION

The brain's nervous system controls human general behaviors; it plays a fundamental role in people's lives, including their decisions, lifestyles, emotions, and judgments. Brain diseases are electrical abnormalities whose severity varies based on several factors, including the patient's age, disorder location in the brain family medical history, and other undefined elements that distinguish one patient from another. For instance, diseases that affect the neural cells include Alzheimer's, epilepsy, schizophrenia (SZ), Parkinson's, *etc*. These disorders have varied symptoms, each affecting a specific area and function of the brain, which is discovered and diagnosed using various tools and tests.

Schizophrenia is a brain disorder burdensome the nerve cells; its symptoms appear obviously between the ages of 16 and 30 (*Huang et al., 2019*; *Li et al., 2023*). The evoked symptoms include delusions, hallucinations, depression, anxiety (*Jauhar, Johnstone & McKenna, 2022*), *etc*. People who have schizophrenia may have an abnormal perspective of the world around them (*Zhu et al., 2022*). Schizophrenia is one of the most severe illnesses that can eradicate the brain's neural system. The WHO organization's official report declared that there are more than 21 million subjects, of which around 1% suffer from mental diseases, all over the world (*Sadeghi et al., 2022*). These symptoms can be utilized to identify the patient's condition into positive and negative (or cognitive) categories; thus, timely detection is considered an essential part of the recovery process for patients (*Aslan & Akin, 2020*). The neuroimaging techniques field provided the knowledge underlying symptoms of schizophrenia and other brain disorders, inspiring and accelerating scientific development to reach new goals in the medical field. The conventional technique for diagnosing ScZ is based entirely on the unique patient's response and the experience psychiatrists have, which makes this process extremely subjective, biased, and time-consuming.

In recent years, electroencephalography devices (EEG) have shown a magnificent contribution to dealing with nervous system diseases and diagnosing them, such as epilepsy (*Al-Azzawi, 2021*; *Al-azzawi et al., 2022*), Alzheimer's disease (*Al-Jumaili et al., 2023*; *Ferdowsi et al., 2024*; *Kim et al., 2024*; *Nour, Senturk & Polat, 2024*), and schizophrenia (*Khare, Bajaj & Acharya, 2023*). EEG was the ideal instrument for capturing the electrical activity of the brain using a variety of electrodes because of its non-invasive nature, which provides a high and complex brain dimensionality that contains a huge amount of data. EEG penetrates and collects the brain's electrical activity, enabling the ability to diagnose illnesses that had hitherto eluded specialists. These recorded signals are then clearly sent to the outside world, which handles them in various ways that support the scientific objective. On the same side, in the medical field, machine learning (ML) is one of the most significant techniques that provides a sophisticated understanding of handling, processing, and analyzing various dataset types such as brain signals, genetic information, and medical images (*Al-Jumaili et al., 2021*; *Al-Jumaili, Duru & Uçan, 2021*). Besides, ML could diagnose, detect, predict, and classify different diseases precisely. ML handles schizophrenia detection in a new computational method compared to traditional methods,

where signals' hidden information is extracted from the data using a feature extraction technique.

Furthermore, the extracted features can reduce the signal's dimensions, quantifying the most related data. In addition, these properties are used with either the frequency domain or the time domain (*Aksöz et al., 2022*; *Gosala et al., 2023*). Subsequently, many types of algorithms are utilized to enhance the precision of the findings of the analysis, diagnostic, and detection procedures. Linear discriminant analysis (LDA), support vector machines (SVM), and K-nearest neighbors (KNN) for instance. Since the precision ratio from previously published studies is generally fairly acceptable, we attempted to significantly improve the accuracy by proposing a method which summarize the most significant proposed methodological contributions in the detection of schizophrenia by EEG signals as follows:

1) To propose a highly effective method for classifying epoched EEG signals of SZ and healthy controls using machine learning algorithms in terms of decreased complexity.
2) To investigate the effects of augmentation techniques using two methods SNR and stretch on classification accuracy.
3) To compare the classification performance of less number of electrode usage with varying epoch lengths of 1-, 2-, and 5-s.

## RELATED WORK

There have been several publications and articles about the classification of schizophrenia; one of these articles was done by *Hartini & Rustam (2021)*, classified the Schizophrenia dataset which had two groups (schizophrenic and non-schizophrenic). RBF and polynomial kernel functions were utilized, and they applied four different k-fold validation methods (3, 5, 7, and 10). The best outcome obtained when K = 10 was 69%. *de Miras et al. (2023)*, to acquire machine learning classifiers for schizophrenia based on resting-state EEG data, they have assessed if machine learning techniques may aid in the diagnosis of the disorder. They have also developed a processing pipeline. They tested five machine learning algorithms: support vector machines (SVM), k-nearest neighbors (kNN), logistic regression (LR), decision trees (DT), random forest (RF), and SVM. SVM produced the best classification results (89%) when it came to separating patients with schizophrenia from healthy subjects. Further, *Hassan, Hussain & Qaisar (2023)*, employed a multi-channel EEG signal dataset in their study to identify schizophrenia. They created a channel selection mechanism based on a thorough performance analysis of the convolutional neural network (CNN) while taking into account the unique EEG channels in various brain regions. To train the classification model, they combined several machine learning (ML) classifiers with CNN. Their results demonstrate that a hybridization of CNN and logistic regression (LR) utilizing three channels—T4, T3, and Cz—achieves 90% and 98% accuracy, respectively. *Siuly et al. (2020)* used EEG signal data with two groups, which are schizophrenia and healthy subjects. They applied empirical mode decomposition to the signals and then applied the Kruskal-Wallis test to select the most significant features.

Then, all features were fed to the SVM to classify them. The highest result achieved was 93.21%. In their research, *Khare et al. (2020)* got 88% with SVM out of the five classification techniques used, including KNN, DA, ensemble method, and decision tree. They applied kurtosis, variance, root mean square, mean, and minimum as feature extraction methods on the EEG signal of the schizophrenia dataset and then used the Kruskal-Wallis test to select the best features. Additionally, *Jain et al. (2022)*, classified the EEG signals into schizophrenia and control groups. Empirical Wavelet Transform has been used in their study to decompose the signals and they applied three different types of entropy (sample entropy, Shannon entropy, and log energy entropy) as feature extraction. Then four techniques were used (support vector machine (SVM), k-nearest neighbor (KNN), linear discriminant (LD), and neural network) with two types of K-folds (5 and 10) to achieve their goal. The best classification accuracy they achieved was 87% using SVM, KNN, and a neural network with k = 5.

Meanwhile, in this study, we proposed a new technique for schizophrenia disorder classification to distinguish schizophrenia patients from healthy people by using EEG signals. The EEG dataset is applied to the band-pass filter as a preprocessing phase, which blocks signals at undesirable frequencies from passing through in preparation for schizophrenia disorder classification. After preprocessing, the filtered EEG signals are split into the delta, theta, alpha, beta, and gamma frequency sub-bands. The suggested method's primary objective is to precisely categorize schizophrenia by extracting the perceptive features of the Fast Fourier Transform, approximate entropy, log energy entropy, Shannon entropy, and kurtosis for every frequency band related to the EEG data signals. Also, each EEG band has been normalized by L2-normalization methods, where the normalization of EEG signals helps accuracy and performance enhancement in conventional ML models. Finally, these features of the EEG signals are fed into the three supervised machine learning classifiers used.

Based on the results obtained in previous studies and according to Table 1, which highlights the researchers' use of various classifiers to classify schizophrenia disorder, we observe that different datasets with various techniques had been used, and the results were differentiated and did not meet expectations. Therefore, we propose a method that has the ability to further increase the accuracy of the diagnosis of schizophrenia.

## MATERIALS AND METHODS

In the following paragraphs, this article will present the impact of the window size and the band-pass filter on classifying schizophrenia EEG signals using four traditional ML models. For the classification technique, firstly, we discuss the dataset availability, the preprocessing technique, and finally, the ML models used for SZ classification.

### Dataset

The dataset used in our approach is a publicly available EEG dataset (*Olejarczyk & Wojciech, 2017*). The signals employed contain 28 subjects, 14 from each group: patients with paranoid schizophrenia and healthy controls under resting state with eyes closed condition.

**Table 1 Studies classified schizophrenia patients and healthy controls using different techniques.**

| Ref. | Year | Features | Electrode no. | Number of subjects | | Classifiers | Acc % |
|------|------|----------|---------------|---------|---------|-------------|-------|
| | | | | Patients | Control | | |
| *Devia et al. (2019)* | 2019 | Event-related potentials | 32 | 11 | 9 | LDA | 71 |
| *Phang et al. (2019a)* | 2019 | CNN | 16 | 45 | 39 | SVM | 92.87 |
| *Torres Naira & Del Alamo (2020)* | 2019 | PCC | 16 | 45 | 39 | CNN | 90 |
| *Rajesh & Kumar (2021)* | 2021 | SLBP | 16 | 45 | 39 | Logitboost | 91.66 |
| *de Miras et al. (2023)* | 2023 | Multiple features techniques | 31 | 11 | 20 | KNN | 87 |
| *Kim et al. (2021)* | 2021 | Multiple features techniques | 19 | 14 | 14 | SVM | 75.64 |
| *Ko & Yang (2022)* | 2022 | Gramian angular field | 9 | 49 | 32 | VGGNet | 93.20 |
| *Buettner et al. (2019)* | 2019 | FFT | 19 | 14 | 14 | Random forest | 80 |
| *WeiKoh et al. (2022)* | 2022 | Local configuration pattern | 19 | 14 | 14 | KNN | 97.20 |
| *Lillo, Mora & Lucero (2022)* | 2022 | Micro-states | 19 | 14 | 14 | CNN | 93 |
| *ArivuSelvan & Moorthy (2020)* | 2020 | Thalamic | NA | 115 | 76 | ANN | 83 |
| *Park et al. (2020)* | 2020 | Hippocampus | NA | 86 | 66 | LR, AB, XGBoost, and SVM | 80.4 |
| *Santos Febles et al. (2022)* | 2020 | NA | 64 | 54 | 54 | Multi-kernel SVM | 83 |
| *De Rosa et al. (2022)* | 2022 | Hippocampus | NA | 20 | 20 | RF | 95 |
| *Chin et al. (2018)* | 2018 | NA | NA | 141 | 71 | SVM | 92 |
| *Sutcubasi et al. (2019)* | 2019 | NA | NA | 93 | 23 | ANN | 81.25 |
| *Vyškovský, Schwarz & Kašpárek (2019)* | 2019 | NA | NA | 52 | 52 | MLP | 73.12 |
| *Baygin et al. (2023)* | 2023 | TQWT | NA | NA | NA | KNN | 99.20 |
| *Khare, Bajaj & Acharya (2023)* | 2023 | Margenau–Hill time-frequency distribution | 19 | 14 | 14 | CNN | 97.4 |
| | | | 16 | 45 | 39 | | 99 |
| | | | 64 | 49 | 32 | | 96 |
| *Khare, Bajaj & Acharya (2023)* | 2021 | SPWVD | 64 | 49 | 32 | CNN | 93.36 |

Table 2 illustrates the EEG dataset details, where the montage was performed using a standard 10–20 system and the dataset was compiled using the following 19 EEG channels: Fp1, Fp2, F7, F3, Fz, F4, F8, T3, C3, Cz, C4, T4, T5, P3, Pz, P4, T6, O1, and O2 (*Sazgar et al., 2019*).

## Processing and feature extraction

EEG signals may contain a variety of noises that have a noticeable impact on classification accuracy; therefore, preprocessing is an essential step in diminishing noise (artifacts). Thus, various filters can be applied to enhance the signal quality. Initially, we used a frequency filter to decompose the EEG signal and eliminate redundant frequencies. The bandpass filter divides data into five frequency sub-bands: delta rhythm (0.1–4 Hz), theta rhythm (5–9 Hz), alpha rhythm (10–14 Hz), beta rhythm (15–31 Hz), and gamma rhythm (32–100 Hz), by allowing only certain frequencies to pass through readily. This helps to filter out frequencies that are too high or too low, where there are a lot of high-frequency

**Peer**J Computer Science

**Table 2** The details of the datasets used.

| Features | Values |
| --- | --- |
| Total subjects | 28 |
| HC | 14 |
| SZ | 14 |
| Males (SZ) | 7 |
| Females (SZ) | 7 |
| Males (HC) | 7 |
| Females (HC) | 7 |
| Mean age (SZ) | 28.1 ± 3.7 years |
| Mean age (HC) | 27.75 ± 3.15 years |
| Mean age (Male SZ) | 27.9 ± 3.3 years |
| Mean age (Male HC) | 26.8 ± 2.9 years |
| Mean age (Female SZ) | 28.3 ± 4.1 years |
| Mean age (Female HC) | 28.7 ± 3.4 years |
| EEG segment | 15 min |
| No. of segments | 21,702 |
| No. of segments without artefacts | 30 |
| No. of channels | 19 |
| Sampling Freq. (Hz) | 250 |

and noise components in the original brain data, and filtering functions help to identify the EEG's detailed information.

These settings achieve the best possible compromise between reducing artifacts that are outside the relevant range of human EEG signals and preventing the development of new artifacts.

We applied the bandpass filter to the EEG data before it was segmented into specific window length. Figure 1 shows the results before and after applying the band-pass filter.

Due to the limited number of subjects, we applied two augmented methods to the dataset involved: AWGN and Stretch to increase the size of the data by generating more training data, which positively effect intensive classifier learning. First, we applied AWGN which is a kind of noise that is utilized to add random noise to the EEG by adding Gaussian noise (signal-to-noise ratio (SNR)). Normally, the addition of SNR is in dB, and we applied a 10dB value with Signal power (measured). Equation (1) illustrates the mathematical representation of AWGN.

$$SNR = 10.\log_{10}\left(\frac{P_{signal}}{P_{noise}}\right). \tag{1}$$

Second, we applied a stretching method to augment the signal, which in turn increases the training of the classifier. In addition, we used it to examine our proposed approach to classifying schizophrenia subjects. We performed the stretch method on the x-axis with a 20% time stretch for the original signal and employed it for the 19 channels.
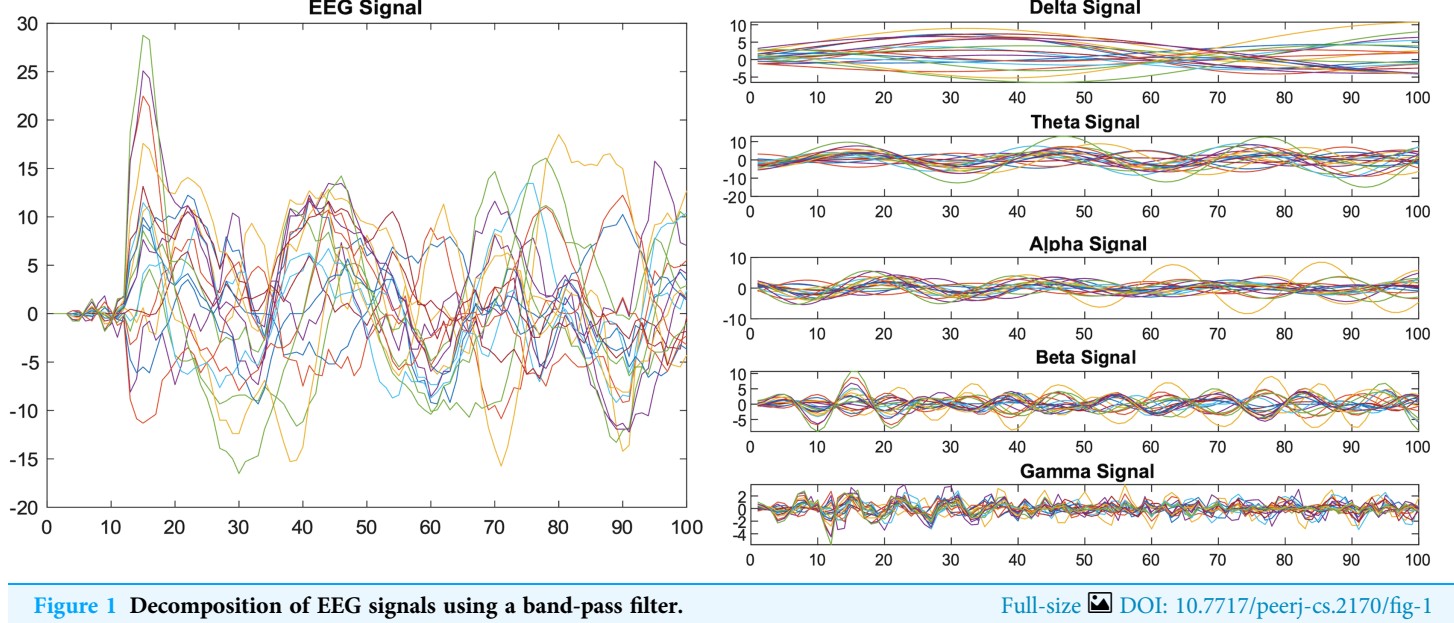

**Figure 1** Decomposition of EEG signals using a band-pass filter.

Due to the biophysical properties such as conductivity differences of brain, skull and scalp tissues, it is not straightforward to investigate the cortical activity from the measured scalp potentials with a few electrodes. On the other hand, increasing the number of scalp electrodes requires more attention to decrease the impedance which is one of the crucial signal quality issues. Thus, the determination of the best-performing electrodes may allow for faster and cost-effective data collection. For this reason, we used different scenarios by reducing features using the MRMR feature selection technique. The MRMR algorithm is a sequential feature selection technique that locates an ideal set of characteristics that are both mutually and maximally different (*Ding & Peng, 2005*). The MRMR maximizes the feature set's relevance to the response variable and minimizes its redundancy.

Since there is not a single, accepted rule for feature limitations, the number of features is open to debate. We discussed two feature reduction scenarios to examine our proposed model, which are eight, and five the best features.

In terms of window length, there are no standard criteria used by researchers to segment signals into a specific window size, while many studies have been conducted using different epochs window sizes (*Kim et al., 2021*; *Sun et al., 2021*; *Vázquez, Maghsoudi & Mariño, 2021*; *Agarwal & Singhal, 2023*; *de Miras et al., 2023*; *Ranjan, Sahana & Bhandari, 2024*). In order to investigate which time window is optimal, we used three different epoch window sizes for classification, and since the brain signals contain many frequencies, the choice of window size is crucial in affecting the final results: Therefore, a bandpass filter was used. The distribution range of the signal was between (0.1 to 100), where low frequencies (such as delta and theta) require a large window size. In contrast, higher frequencies (eg, beta, gamma) require a small window size. Therefore, the choices of the epoch sizes were (1, 2, and 3 s) to capture all the frequencies and evaluate the proposed

method if it has the ability to obtain promising results. Also, in terms of computational efficiency, we evaluate the window size effect on feature extraction and the final results of the classifier, besides the time duration needed to predict speed and training time for the same classifier.

Bear in mind that determining classification accuracy depends on the ability to extract features from EEG signals, which is not just a difficult phase but also a critical and challenging step in the classification process. For schizophrenia, EEG patterns can be extracted using both time-domain and frequency-domain feature techniques. Time-domain feature extraction techniques use changes in signal time series to analyze EEG data, whereas non-linear analysis techniques have recently been widely used to analyze EEG data. As the next step to the preprocessing phase, various methods are used with EEG signals, such as fuzzy entropy, permutation entropy, symbolic dynamics-based entropy, sample entropy, time-frequency distributions, wavelet transform, and eigenvector methods to extract features.

In this study, six extraction methods were calculated for each window epoch to extract the hidden features. We implemented two cases: features implemented with the band-pass filter and features implemented without. Specifically, FFT, log energy entropy, kurtosis, and Shannon entropy were implemented with the band-pass filter; in contrast, approximate entropy was applied in both cases with the band-pass filter and without.

### Fast Fourier transform

Fast Fourier transform (FFT) has changed the world, is one of the most important algorithms developed of all time, and is the enabling piece of technology in most digital communication, audio, image compression, and signals. The real reason for depending on FFT is its quick and effective method of denoising data. The FFT feature has been implemented on the SZ EEG signals to convert the time domain to the frequency domain (*Sedik, Marey & Mostafa, 2023*). It helps measure the power spectrum of the data from the frequency band. Equation (2) below illustrates the mathematical formulation of FFT, where X(K) means the Fourier coefficient of x(n), and the odd n and even n are compatible with odd numbers and even numbers with the frequency of K, respectively.

$$X(K) = \sum_{n=0}^{N-1} X[n] W_N^{kn} = \sum_{n\ even} x(n)nw_N^{kn} + \sum_{n\ odd} x(n)w_N^{kn} \tag{2}$$
$$K = 0, 1 \ldots \ldots, N-1,$$

### Approximate entropy

Entropy is the most frequently used feature to measure time-domain features and is also widely used in disease detection. Approximate entropy (ApEn) is defined as a measurement of the regularity or randomness of data in a time series and is used for short-length data due to its lower sensitivity to noise. Equation (3) shows the mathematical formula, where r stands for the similarity criterion, e stands for (the length of the data segment being compared), and N stands for (the length of data).

$$ApEn(E, r, N) = \frac{1}{(N - e + 1)} \sum_{i-1}^{N-e+1} \log \; C_i^e(r) - \frac{1}{N - e} \sum_{i=1}^{N-e} \log \, C_i^{e+1} \,(r). \tag{3}$$

### Shannon entropy

Shannon and Weaver established entropy for information theory, which characterizes a signal's complexity, irregularity, uncertainty, or unpredictability. ShnEn is a time-domain complexity metric that does not rely on the signal spectrum. A discrete signal's ShnEn may be computed as Eq. (4):

$$Hsh = - \sum_{i=1}^{Na} Pi \, log \, Pi. \tag{4}$$

Na is the total number of amplitude values in the signal with the I range, and pi is the probability of the signal with the ai amplitude.

In practice, instead of calculating all amplitude values, the probability density function of the signal is predicted using the histogram. Signal amplitudes are split into k bins to generate the normalized form of Hsh, and entropy is computed as Eq. (5):

$$ShEn = \frac{HSh}{og \, k} \tag{5}$$

### Log energy entropy

The ShnEn and log energy entropy (LogEn) are calculated using the entropy-based wavelet packet decomposition proposed by Coifman and Wickerhauser. Entropy is utilized as the feature method because it can be used to determine how random the information is *Prasanna, George & Subathra (2024)*. LogEn, which is represented by Eq. (6), is a commonly used metric in signal processing that can extract relevant information:

$$E = \sum_n log \left( w_{i,j}^{n2} \right) \tag{6}$$

where $\left( w_{i,j}^{n2} \right)$ are the computed WPD coefficients.

### Kurtosis

The word kurtosis, which means 'tailedness', comes from the Greek word 'kyrtos' or 'kurtos' (*The MathWorks, Inc., 2022*). It is considered one of the shape measurements and is historically defined as the peakedness of a distribution. Nowadays, it is just clarifying the tail extremity, which means either existing outliers (for the sample kurtosis) or the propensity to produce outliers (for the kurtosis of a probability distribution). A normal distribution will have a kurtosis of three, called mesokurtic, while a distribution of more than three is called leptokurtic and called platykurtic, which is less than three. Kurtosis has been used to find the normality of data for statistical analysis from range 1 to infinity throughout the below Eq. (7) (*The MathWorks, Inc., 2022*).

$$\kappa = \frac{E(x - \mu)^4}{\sigma^4} \tag{7}$$

After the feature extraction step, and due to the features extracted having different scales, normalization is important to ensure the feature will be in the same range, which can improve training speed and performance. The normalized features are input for the classification: SVM, KNN, QDA, and EC. FFT, ApEn, LogEn, ShnEn, and kurtosis are all computed using a MATLAB routine.

## Normalization

Standardization and normalization are the two most common feature scaling methods in machine learning. Normalization is regarded as a crucial phase in the production of data. Generally, normalization is the most widely used method with data in linear transformations. Feature scaling, whose values range from 0 to 1, makes comparing data easier. Therefore, we utilized the L2 normalization method in our approach, and the outcomes served as inputs for the next stage, classification, which raised and improved our accuracy results.

## Classification

The classification of individuals with schizophrenia was done using the following four classifiers widely used in classifying EEG signal datasets. The baseline techniques were employed as part of the proposed ML method described in this section: SVM, KNN, QDA, and EC. The following section will give a quick overview of each of these approaches.

### Support vector machine

In recent years, many articles have used SVM to classify various datasets. The most critical component in SVM is the hyperplane, which affects the accuracy of the result based on the optimal hyperplane used to classify the dataset into different groups. Furthermore, the class boundaries are subject to finding the ultra-optimal level, which removes some irrelevant data from the training data set to reduce classification error. SVM employs a variety of kernels, including Polynomial, Gaussian, Radial Basis Function (RBF), Laplace RBF, Sigmoid, and Anove RBF, *etc.*, which are named a (kernel trick). However, as shown in the mathematical expression in Eq. (8), the two essential parameters are W and (Z-B), where W represents the transport vector and B is the displacement of that vector. The distance from Z to the hyperplane is:

$$D(z) = W \bullet Z - B, \ where \ Z \in \begin{cases} A \ if \ D(z) > 0 \\ B \ if \ D(z) < 0 \end{cases}. \tag{8}$$

### K-nearest neighbors

KNN is a non-parametric classification method that uses a distance check to discover the classifier's closest neighbors. The Euclidean distance equation is used to calculate distance, and the mathematical formula is presented below in Eq. (9). To classify a certain (X), the classifier check measures the distance between the (X) and the other data during the training phase. It assigns it a specific label (K) indicating its class, and KNN will do the same with all data until all data is.

$$dis(x1, x2) = \sum_{i=1}^{n} \sqrt{(x1i - x2i)^2}. \tag{9}$$

### Quadratic discriminant analysis

Quadratic discriminant analysis (QDA) is a machine learning and statistical classification classifier that uses quadric surfaces to classify two or more classes of different types of information. It is a more appropriate form than a linear classifier. QDA, in particular, employs a Gaussian distribution for each class. As demonstrated by mathematical Eq. (10). $P(x \mid y = k)$ is modeled as a multivariate Gaussian distribution with density, and d is the number of features.

Figure 2 illustrates the pipeline for the categorization method applied in this research using the entropy family, FFT, and kurtosis.

$$P(x \mid y = k) = \frac{1}{(2\pi)^{\frac{d}{2}} |\sum k|^{\frac{1}{2}}} \exp\left(-\frac{1}{2}(x - \mu_k)^t \sum_k^{-1} (x - \mu_k)\right). \tag{10}$$

### Ensemble classifier

It is a machine learning method that combines more than one model to provide an accurate classification. There are different types of algorithms used such as decision trees, neural networks, support vector machines, or k-nearest neighbors. It relies on each model's advantages to provide an outcome that is more robust and durable than any of the models alone. Ensemble classifier (EC) can handle the complexity of a multidimensional dataset and reduce the variance and bias of classification results, by overcoming the weak points of each classifier (*Ranjbari et al., 2021*).

### Evaluation metrics

In this section, we explain the method that we used to validate each model independently. We used multiple forms of performance rating scales by using a confusion matrix which is one of the most used methods to examine, evaluate, and represent the classifier efficacy. The results were obtained through the classifiers using four sorts of outcomes, namely true positive (TP), true negative (TN), false positive (FP), and false negative (FN). Table 3 shows the formulas used to compute the assessment metrics: accuracy, sensitivity, specificity, precision, negative predictive value (NPV), F1-Score, and Matthew's correlation coefficient (MCC) (*Chicco & Jurman, 2023*).

## RESULTS

This section presents the results of our proposed method. This article investigated the role of the window epoch size and the impact of the band-pass filter in schizophrenia EEG signal classification using the database from (*Olejarczyk & Wojciech, 2017*). First, the used dataset had two classes: 14 patients with schizophrenia condition and 14 with healthy conditions, and the dataset was available and could be accessed. The recording signals were

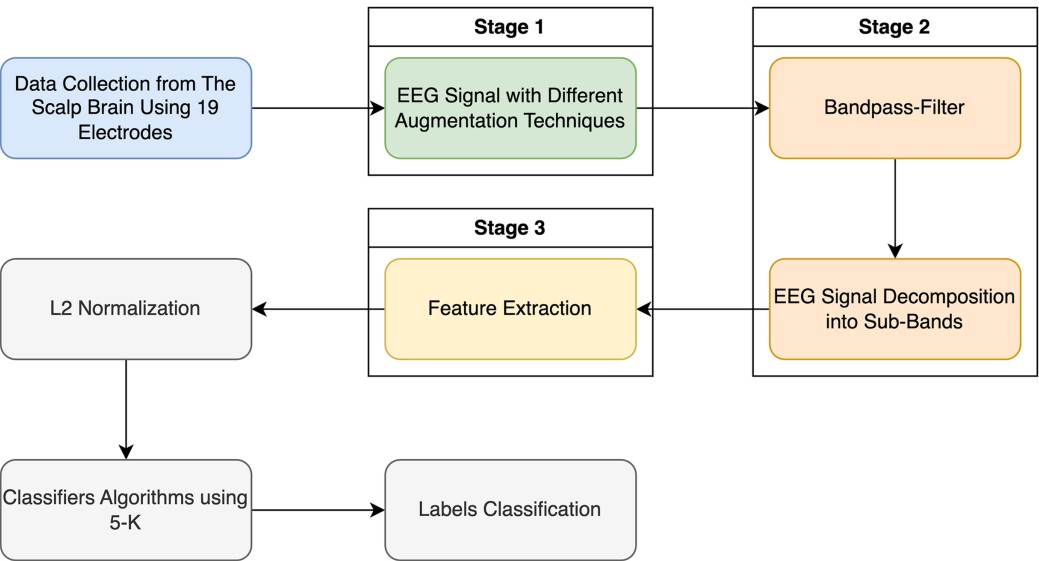

**Figure 2  Classification EEG signal data with schizophrenia disorder in two labels: schizophrenia and healthy control.**               

**Table 3  Evaluation metrics.**

$Accuracy = (TP + TN)/(TP + FP + TN + FN)$

$Sensitivity = TP/(TP + FN)$

$Specificity = TN/(TN + FN)$

$Precision\ (PPV) = TP/(TP + FP)$

Negative Predictive Value $(NPV) = TN/(TN + FN)$

$F1 - score = (2 * TP)/(2 * TP + FP + FN)$

$MCC = ((TP * TN) - (FP * FN))/\sqrt{((TP + FP)(TP + FN)(TN + FP)(TN + FN))}$

collected from 19 channels, and all the outcomes were evaluated according to the frequencies of 250 Hz. In addition, we investigated the impact of the band-pass filter on the signals that had been recorded by EEG, which affected the performance of the models. In addition to the traditional techniques, we used data-augmented methods (AGW and stretched signals) and the MRMR features selection method using the eight and five best features.

Then, we looked at the effectiveness of six feature extraction techniques (FFT, ApEn, LogEn, ShnEn, and kurtosis) side by side with SVMs, KNNs, QDA, and EC.

After extracting the attributes, we normalized the data using the L2-normalization method, and then the algorithms were trained using three epoch sizes (1, 2, and 5 s) to assess more effectively. This produced three scenarios using machine learning algorithms for classifying schizophrenia disorder according to each epoch's size.

## One-second epoch size

In the first scenario, we used a 1-s epoch window size and relied on the performance of the six features. These were used as an effective tool for EEG signal analysis and have been applied to diagnose neurological diseases.

SVM and EC outperformed the other two classifiers with 99% accuracy when implementing LogEn; in contrast, KNN, and QDA obtained 98%, and 96% accuracy values, respectively. Table S1 presents the actual and predicted values of the confusion matrix for the four classifiers, while Table 4 demonstrates the calculated parameters: sensitivity, specificity, precision, NPV, F1-Score, and MCC. The results obtained with the highest accuracy for the above parameters were 99% for SVM and EC. According to all the above results, KNN was in second place, while QDA came last.

Moreover, Fig. S1 illustrates the ROC curves obtained, particularly the 1-s epoch size with pure dataset (without augmented or reduced features), using the six features' methods, including the ApEn with and without Bandpass-Filter for the SZ dataset. These diagrams visually represent the classification performance, showing how the four classifiers used on the SZ dataset are between the true positive rate (sensitivity) and the false positive rate (specificity).

For utilizing the SNR signal as a dataset applied to the feature extraction methods, Table S2 represents the confusion matrices for all classifiers. When using the results obtained from the SNR augmented method, they were all ideal using LogEn feature with Band-pass filter. Further, Table 5 illustrates the measured parameters for classifiers and shows 99% for all. Moreover, the prediction speed and training time were fairly acceptable.

Figure S2 illustrates the ROC curves obtained, particularly the 1-s epoch size and applying the SNR augmented method on the dataset, using the six features' 'methods'. These graphs visually represent the classification performance, showing how the four classifiers used on the SZ dataset are between the true positive rate (sensitivity) and the false positive rate (specificity). The effectiveness of an individual classifier is represented by each curve on the ROC graph.

Table S3 shows the confusion matrices for all classifiers when using the Stretch signal as a dataset for feature extraction. When employing the Stretch-augmented technique, all the results were optimal when combined with the LogEn feature and a Band-pass filter. Furthermore, Table 6 provides the measured parameters for classifiers, all accuracies were fairly acceptable for all classifiers. Likewise, the prediction speed and training duration were also adequate.

The ROC curves generated using the six features' methods and the stretch approach applied to the dataset are shown in Fig. S3, with special attention to the 1-s epoch size. The four classifiers employed on the SZ dataset are represented visually in these graphs and the effectiveness of an individual classifier is represented by each curve on the ROC graph.

The latest developments indicate that utilizing EEG with fewer features, possibly even as few as five or eight, could completely transform the technology's usability and affordability. This technique aims to find the optimal features to work with, which saves setup time, and complexity. In the concept of this study, two scenarios are used to reduce the features by

**Table 4 One-second epoch size classification performance results of four classifiers.**

| CT | FE | Evaluation metrics | | | | | | | | |
|----|----|-----|-----|-----|-----|-----|----------|-----|------------------------|---------------------|
| | | Acc | Sen | Spe | Pre | NPV | F1-score | MCC | Prediction speed (sec) | Training time (sec) |
| SVM | FFT | 97 | 97 | 97 | 97 | 98 | 97 | 95 | 11,879.4 | 250.2 |
| KNN | | 95 | 95 | 95 | 95 | 95 | 95 | 90 | 1,196.39 | 108.48 |
| QDA | | 93 | 95 | 93 | 91 | 95 | 93 | 87 | 105,281.1 | 3.769 |
| EC | | 96 | 96 | 97 | 97 | 96 | 96 | 93 | 124.4 | 1,262.4 |
| SVM | ApEn | 82 | 87 | 78 | 78 | 86 | 82 | 65 | 5,295.21 | 875.15 |
| KNN | | 73 | 85 | 63 | 68 | 82 | 75 | 49 | 1,221.87 | 298.98 |
| QDA | | 75 | 98 | 53 | 66 | 97 | 79 | 57 | 63,405.56 | 5.025 |
| EC | | 85 | 81 | 89 | 87 | 84 | 84 | 71 | 676 | 155 |
| SVM | ApEn + Bandpass | 82 | 86 | 77 | 78 | 86 | 82 | 64 | 104,405.87 | 1,453.77 |
| KNN | | 73 | 84 | 62 | 67 | 81 | 75 | 46 | 5,236.17 | 21.4 |
| QDA | | 74 | 98 | 53 | 66 | 97 | 79 | 57 | 674,686.27 | 1.11 |
| EC | | 81 | 74 | 92 | 94 | 70 | 83 | 66 | 14,347.6 | 1,326.4 |
| SVM | Shannon entropy | 87 | 83 | 93 | 94 | 81 | 88 | 76 | 114,062.06 | 3,434.53 |
| KNN | | 82 | 76 | 88 | 89 | 74 | 82 | 64 | 833.21 | 238.26 |
| QDA | | 92 | 92 | 92 | 93 | 90 | 93 | 84 | 94,346.25 | 4.578 |
| EC | | 95 | 94 | 96 | 96 | 95 | 95 | 91 | 28,210.4 | 2,641.6 |
| SVM | Log energy entropy | 99 | 99 | 99 | 99 | 99 | 99 | 99 | 31,065 | 2,370.53 |
| KNN | | 98 | 98 | 99 | 99 | 98 | 99 | 97 | 390.78 | 321.09 |
| QDA | | 96 | 94 | 99 | 99 | 93 | 97 | 93 | 137,986.41 | 5.36 |
| EC | | 99 | 99 | 99 | 99 | 99 | 99 | 98 | 113.1 | 1,280.4 |
| SVM | Kurtosis | 73 | 71 | 75 | 78 | 68 | 74 | 47 | 1,319.60 | 96.82 |
| KNN | | 63 | 78 | 45 | 63 | 63 | 70 | 25 | 7,330.22 | 2,659.32 |
| QDA | | 68 | 59 | 80 | 78 | 61 | 67 | 39 | 117,144.71 | 4.076 |
| EC | | 71 | 76 | 67 | 70 | 73 | 73 | 43 | 12,829.4 | 1,286.9 |

Note:
  Where Classifier Types (CT), Feature Extraction (FE), Accuracy (Acc), Sensitivity (Sen), Specificity (Spe), Precision (Pre).

selecting eight and five best features. Tables S4 and S5 show the results of the confusion matrix obtained by using the features after reducing to five and eight and using these features as inputs to the classifiers. Tables 7 and 8 show the ability of the classifiers to classify with fewer features, as the results obtained were good compared to the results obtained using 19 channels.

Figures S4 and S5 display the ROC curves produced by applying the reduction features strategy to the dataset and the six feature approaches. The 1-s epoch size should be noted. These graphs show the four classifiers used on the SZ dataset visually, with each curve on the ROC graph representing the efficacy of a single classifier.

## Two-second epoch size

The next step of the proposed method is to classify schizophrenia by the selected features with a 2-s epoch window size. In the process of training and testing the EEG data using the four algorithms with the 19-electrode dataset. The confusion matrix of the proposed

**Table 5 One-second epoch size classification performance results with SNR for four classifiers.**

| CT | FE | Evaluation metrics | | | | | | | | |
|----|----|-----|-----|-----|-----|-----|----------|-----|----------------------|--------------------|
| | | Acc | Sen | Spe | Pre | NPV | F1-score | MCC | Prediction speed (sec) | Training time (sec) |
| SVM | FFT | 97 | 97 | 98 | 97 | 98 | 97 | 95 | 18,340.3 | 190 |
| KNN | | 96 | 95 | 96 | 95 | 96 | 95 | 92 | 424.4 | 1,054.4 |
| QDA | | 95 | 93 | 97 | 96 | 94 | 95 | 91 | 32,623.1 | 21.1 |
| EC | | 97 | 96 | 97 | 97 | 97 | 97 | 94 | 31.7 | 5,436.5 |
| SVM | ApEn | 72 | 66 | 78 | 78 | 67 | 71 | 45 | 3,962.8 | 851.7 |
| KNN | | 70 | 63 | 79 | 80 | 61 | 71 | 42 | 4,113.2 | 1,104.4 |
| QDA | | 65 | 62 | 67 | 58 | 70 | 60 | 29 | 281,495 | 1.9 |
| EC | | 69 | 63 | 75 | 73 | 65 | 68 | 39 | 11,821.7 | 2,369.5 |
| SVM | ApEn + Bandpass | 72 | 68 | 77 | 75 | 70 | 71 | 45 | 1,152.3 | 1,688.6 |
| KNN | | 69 | 62 | 78 | 79 | 60 | 70 | 41 | 586.1 | 3,191.7 |
| QDA | | 69 | 61 | 81 | 84 | 57 | 71 | 42 | 39,658.7 | 24.1 |
| EC | | 67 | 64 | 70 | 63 | 71 | 64 | 34 | 5,181.5 | 4,721.9 |
| SVM | Shannon entropy + Bandpass | 90 | 86 | 95 | 94 | 87 | 90 | 81 | 2,894.6 | 821.8 |
| KNN | | 95 | 94 | 95 | 94 | 95 | 94 | 90 | 400.1 | 3,061.6 |
| QDA | | 69 | 59 | 96 | 98 | 45 | 74 | 49 | 60,912.9 | 15.3 |
| EC | | 99 | 99 | 99 | 99 | 99 | 99 | 98 | 38,264.8 | 3,527.2 |
| SVM | Log energy entropy + Bandpass | 99 | 99 | 99 | 99 | 99 | 99 | 99 | 47,392.1 | 98.1 |
| KNN | | 99 | 99 | 99 | 99 | 99 | 99 | 99 | 554.5 | 405.4 |
| QDA | | 99 | 99 | 99 | 99 | 99 | 99 | 99 | 34,577.3 | 24.5 |
| EC | | 99 | 99 | 99 | 99 | 99 | 99 | 99 | 40 | 5,756.9 |
| SVM | Kurtosis + Bandpass | 67 | 63 | 70 | 65 | 68 | 64 | 34 | 826.5 | 1,924.4 |
| KNN | | 62 | 59 | 64 | 54 | 68 | 56 | 23 | 785.2 | 3,338 |
| QDA | | 59 | 58 | 59 | 34 | 79 | 43 | 16 | 32,784.7 | 29.2 |
| EC | | 63 | 57 | 72 | 74 | 54 | 65 | 30 | 30,899.4 | 4,791.8 |

**Note:**
  Where Classifier Types (CT), Feature Extraction (FE), Accuracy (Acc), Sensitivity (Sen), Specificity (Spe), Precision (Pre).

method shown in Table S6, summarizes the actual and predicted values obtained from the experiments. Whereas the results are shown in Table 9, which compares the accuracy resulting from the classification of schizophrenia in the proposed model with other learning models. These results show that the proposed method has an average accuracy of 99% in classifying healthy and schizophrenia subjects and has misclassified only one of the test samples. On the other hand, the prediction speed with 2-s epoch for both higher classifiers (EC and SVM) were only 204.5921 and 42,483.20 s. Again, the SVM and EC classifiers got identical accuracy utilizing the Log Energy Entropy feature, and compared with the other metrics values, KNN came in second, and QDA came in last.

Although SVM and EC produce superior ROC values for SZ, KNN often performs better in class distinction than SVM. In Fig. S6, when comparing SVM and EC to other classifier techniques, QDA performs worse in the data because its ROC curve is often lower.

**Table 6 One-second epoch size classification performance results with stretch for four classifiers.**

| CT | FE | Evaluation metrics | | | | | | | | |
|----|----|-----|-----|-----|-----|-----|----------|-----|-----------------------|-------------------|
| | | Acc | Sen | Spe | Pre | NPV | F1-score | MCC | Prediction speed (sec) | Training time (sec) |
| SVM | FFT | 97 | 97 | 97 | 97 | 97 | 97 | 94 | 14,815.9 | 336.8 |
| KNN | | 96 | 96 | 95 | 95 | 96 | 95 | 92 | 222.5 | 2,293.3 |
| QDA | | 93 | 90 | 95 | 95 | 90 | 92 | 86 | 42,727.4 | 27.2 |
| EC | | 96 | 95 | 97 | 97 | 95 | 96 | 93 | 30.2 | 7,720.7 |
| SVM | ApEn | 92 | 92 | 92 | 90 | 94 | 91 | 84 | 3,580.1 | 781.1 |
| KNN | | 90 | 91 | 89 | 87 | 93 | 89 | 81 | 1,828.3 | 1,965.6 |
| QDA | | 81 | 84 | 87 | 86 | 85 | 85 | 72 | 350,875.9 | 3.8 |
| EC | | 91 | 89 | 92 | 90 | 91 | 90 | 81 | 507.8 | 2,325.6 |
| SVM | ApEn + Bandpass | 79 | 72 | 87 | 87 | 72 | 79 | 60 | 973 | 1,948.4 |
| KNN | | 71 | 61 | 93 | 95 | 51 | 75 | 50 | 297.5 | 3,079.9 |
| QDA | | 71 | 61 | 97 | 98 | 49 | 75 | 53 | 42,705 | 32.3 |
| EC | | 77 | 67 | 92 | 93 | 63 | 78 | 58 | 12 | 11,851.3 |
| SVM | Shannon entropy + Bandpass | 87 | 80 | 95 | 94 | 80 | 86 | 75 | 886.2 | 2,874.4 |
| KNN | | 93 | 92 | 95 | 94 | 93 | 933 | 87 | 254.5 | 4,914.9 |
| QDA | | 63 | 55 | 96 | 98 | 35 | 71 | 41 | 57,873.2 | 23.5 |
| EC | | 97 | 96 | 98 | 98 | 97 | 97 | 95 | 23,228.9 | 6,893.9 |
| SVM | Log energy entropy + Bandpass | 99 | 99 | 99 | 99 | 99 | 99 | 98 | 20,876.8 | 566.4 |
| KNN | | 98 | 98 | 99 | 99 | 98 | 98 | 97 | 256.5 | 30,776.2 |
| QDA | | 96 | 92 | 99 | 99 | 93 | 96 | 92 | 31,882.6 | 26.8 |
| EC | | 99 | 99 | 99 | 99 | 99 | 99 | 98 | 11.1 | 38,298.1 |
| SVM | Kurtosis + Bandpass | 73 | 67 | 82 | 82 | 66 | 74 | 49 | 499.8 | 2,660.4 |
| KNN | | 69 | 65 | 72 | 67 | 70 | 66 | 38 | 330.8 | 4,418.7 |
| QDA | | 49 | 47 | 87 | 98 | 100 | 63 | 17 | 38,595 | 30.3 |
| EC | | 73 | 66 | 83 | 83 | 65 | 74 | 49 | 10,616.1 | 7,548.9 |

**Note:**
Where Classifier Types (CT), Feature Extraction (FE), Accuracy (Acc), Sensitivity (Sen), Specificity (Spe), Precision (Pre).

Then, the 2-s epoch window size was applied to the data that was augmented by using the SNR method. The confusion matrix results of our technique are clarified in Table S7, whereas the performance parameters recorded greatly varying rates with the six features utilized, as illustrated in Table 10. In contrast, the four classifiers using log energy entropy with a bandpass filter obtained equal and ideal values, which is a result of the 2-s epoch window size that was used.

For visual representation, the ROC is considered one of the methods used to check the performance of classifiers when comparing different classes. Figure S7 illustrates the ROC curves produced by applying the SNR approach to the dataset and the six attributes. The graphs show the four classifiers used on the SZ dataset visually, with each curve on the ROC graph representing the success rate of a single classifier.

In addition to the SNR technique, the dataset underwent the stretch augmentation method, a technique for enhancing classification accuracy by diversifying the data's plausible changes. It was a way to tackle the small data issue and has shown an encouraging

**Table 7 One-second epoch size classification performance results with five electrodes for four classifiers.**

| CT | FE | Evaluation metrics | | | | | | | | |
|----|----|-----|-----|-----|-----|-----|----------|-----|--------------------------|---------------------|
| | | Acc | Sen | Spe | Pre | NPV | F1-score | MCC | Prediction speed (sec) | Training time (sec) |
| SVM | FFT | 92 | 90 | 94 | 93.7 | 90.8 | 91.9 | 84 | 21,765 | 501.3 |
| KNN | | 90.8 | 86.7 | 95 | 94.6 | 87.6 | 90.5 | 82 | 1,350.7 | 471.2 |
| QDA | | 81.7 | 71.8 | 92.9 | 92 | 74 | 80 | 65.6 | 186,543 | 2.5 |
| ENS | | 91.9 | 91 | 92.7 | 93 | 90.6 | 92 | 83.8 | 98.2 | 1,870 |
| SVM | ApEn | 74 | 65.5 | 84.7 | 84 | 66 | 73.7 | 50 | 3,980.7 | 398.3 |
| KNN | | 72.6 | 68.8 | 77 | 78.6 | 67 | 73 | 45.9 | 1,492.1 | 740.1 |
| QDA | | 67.5 | 68 | 65 | 87.6 | 36 | 76.6 | 28 | 312,847.1 | 22.1 |
| ENS | | 72 | 64.5 | 81.5 | 80 | 66.6 | 71 | 46 | 17,681.4 | 727.6 |
| SVM | ApEn + Bandpass | 75.6 | 68 | 85 | 85 | 68 | 75.6 | 53 | 4,978.5 | 786.3 |
| KNN | | 67.8 | 70 | 57.7 | 88 | 29.6 | 78 | 22 | 2,650 | 2,590.1 |
| QDA | | 69.7 | 62 | 95.5 | 97.8 | 42.9 | 75.9 | 48 | 243,101.9 | 2.9 |
| ENS | | 73.7 | 67 | 88 | 92.9 | 54 | 78 | 51 | 51,469.9 | 1,322.8 |
| SVM | Shannon entropy+ Bandpass | 71.9 | 68.6 | 84 | 94 | 41.9 | 79 | 43.6 | 4,299 | 2,598.6 |
| KNN | | 90.9 | 85.7 | 96.6 | 96.5 | 86 | 90.8 | 82 | 1,987.9 | 2,655.1 |
| QDA | | 58.5 | 42.5 | 87 | 85 | 45.9 | 56.8 | 30 | 220,989 | 13.9 |
| ENS | | 96.8 | 95.7 | 97.9 | 97.6 | 96 | 96.7 | 93.7 | 41,987.9 | 3,001.8 |
| SVM | Log energy entropy + Bandpass | 97.7 | 96.6 | 98.9 | 99 | 96 | 97.8 | 95.5 | 35,987.3 | 26,932 |
| KNN | | 97 | 94.7 | 99.6 | 99.6 | 95 | 97 | 94.5 | 1,150.7 | 1,100.9 |
| QDA | | 91.9 | 85.6 | 99 | 98.9 | 85.9 | 91.8 | 84.8 | 67,542.9 | 22.9 |
| ENS | | 98.9 | 98.8 | 99 | 98.8 | 99 | 98.8 | 97.9 | 89.9 | 26,987 |
| SVM | Kurtosis + Bandpass | 67.5 | 64.6 | 73 | 82.7 | 50.9 | 72.6 | 35.6 | 4,562 | 968.9 |
| KNN | | 63 | 50 | 81.9 | 79.8 | 53.7 | 61.7 | 32.9 | 2,879.9 | 1,192.3 |
| QDA | | 48.8 | 48 | 59 | 93.6 | 8 | 63.5 | 3 | 296,150 | 2 |
| ENS | | 68.9 | 62.5 | 85.5 | 91.7 | 47 | 74 | 43 | 43,617.9 | 2,000 |

Note:
Where Classifier Types (CT), Feature Extraction (FE), Accuracy (Acc), Sensitivity (Sen), Specificity (Spe), Precision (Pre).

result. The Confusion Matrix results are shown in Table S8. Here is a breakdown of how well the augmented method performed: The accuracy of this technique, which combines ML models and feature extraction methods, is determined by the percentage of successful classifications, as shown in Table 11, which illustrates the performance of the classifiers. The accuracies varied for the six features; in contrast, the LogEn feature gained with SVM, KNN, and EC 99%. However, with an accuracy of 97%, the QDA also performs admirably.

The ROC curves obtained *via* the SNR technique on the dataset and the six characteristics are shown in Fig. S8. The four classifiers utilized on the SZ dataset can be seen using such graphs, where each curve on the ROC graph denotes the achievement efficiency of a specific classifier.

Features reduction was applied to the dataset to investigate the possibility of achieving maximum accuracy with a minimum feature number. We applied a 2-s epoch window size to the dataset with feature reduction, which sped up computational time and improved the overall efficiency of EEG testing. This procedure was done in two different cases: the

**Table 8 One-second epoch size classification performance results with eight electrodes for four classifiers.**

| CT | FE | Evaluation metrics | | | | | | | | |
|---|---|---|---|---|---|---|---|---|---|---|
| | | Acc | Sen | Spe | Pre | NPV | F1-score | MCC | Prediction speed (sec) | Training time (sec) |
| SVM | FFT | 95 | 94.5 | 96 | 96.6 | 94 | 95.5 | 90.8 | 3,780.4 | 209.1 |
| KNN | | 94.9 | 95 | 94 | 95 | 94 | 95 | 89.6 | 870 | 554 |
| QDA | | 90.7 | 88 | 92.9 | 92.6 | 88.9 | 90 | 81.5 | 153,476.1 | 15.7 |
| ENS | | 95.6 | 93.6 | 97.8 | 97.9 | 93 | 95.7 | 91 | 66.9 | 2,867 |
| SVM | ApEn | 82 | 77 | 87 | 86.7 | 78 | 81.7 | 64.6 | 6,850.8 | 459 |
| KNN | | 81.8 | 79.7 | 83.9 | 82.8 | 80.9 | 81 | 63.8 | 13,654 | 550 |
| QDA | | 69.8 | 70.9 | 66 | 87.8 | 39.6 | 78 | 31.8 | 269,483.9 | 27.9 |
| ENS | | 80 | 79 | 81.8 | 88.5 | 68.9 | 83.6 | 59 | 27,691.1 | 659 |
| SVM | ApEn + Bandpass | 76.7 | 84.7 | 46.5 | 85.6 | 44.8 | 85 | 30.8 | 3,547.9 | 12.3 |
| KNN | | 70.6 | 68.7 | 73 | 78.9 | 61.8 | 73 | 41 | 889.3 | 1,679.3 |
| QDA | | 68.9 | 88.9 | 61 | 46.9 | 93 | 61 | 45 | 91,734.5 | 27 |
| ENS | | 75.8 | 81 | 57.5 | 86 | 48.7 | 83.7 | 36.9 | 55.9 | 3,879.3 |
| SVM | Shannon entropy + Bandpass | 80.8 | 78 | 83.9 | 86 | 75 | 82 | 61.8 | 7,132.8 | 845.1 |
| KNN | | 94.8 | 93.8 | 95.8 | 95.7 | 94 | 94.8 | 89.7 | 2,029.2 | 1,590.9 |
| QDA | | 59.9 | 51 | 97.7 | 99 | 31 | 67.7 | 38 | 16,573.8 | 2.8 |
| ENS | | 97.8 | 96.6 | 98.9 | 98.8 | 97 | 97.7 | 95.7 | 19,831.3 | 3,981.4 |
| SVM | Log energy entropy + Bandpass | 98.6 | 99 | 97.7 | 97.9 | 99 | 98.6 | 97 | 65,741.8 | 155.2 |
| KNN | | 98.9 | 98.9 | 98.9 | 98.8 | 98.9 | 98.9 | 97.8 | 2,803.9 | 380 |
| QDA | | 94.8 | 90.5 | 99.5 | 99.5 | 90.5 | 94.8 | 90 | 166,790.1 | 4 |
| ENS | | 98.9 | 98.9 | 98.9 | 99 | 98.8 | 99 | 97.9 | 55.3 | 2,500.8 |
| SVM | Kurtosis + Bandpass | 68.6 | 62.6 | 76.5 | 77.9 | 60.7 | 69 | 38.9 | 2,980.6 | 2,691 |
| KNN | | 63.7 | 59 | 68 | 67 | 61 | 63 | 27.9 | 1,877.8 | 1,190.1 |
| QDA | | 49.8 | 47.7 | 90.5 | 98.9 | 8 | 64 | 16.5 | 124,531.5 | 5.9 |
| ENS | | 68.7 | 65 | 74 | 80.7 | 56.5 | 72 | 38.5 | 12,671.9 | 2,098.1 |

**Note:**
    Where Classifier Types (CT), Feature Extraction (FE), Accuracy (Acc), Sensitivity (Sen), Specificity (Spe), Precision (Pre).

highest five and eight features. Tables S9 and S10 show the results obtained using the confusion matrix, which will show the performance of the classifiers with four outcomes: true positive, true negative, false positive, and false negative. As well as the accuracy shown in Tables 12 and 13 of all classifiers and feature extraction methods. The highest results were obtained in both cases (five and eight features) using the features of Log Energy Entropy. While FFT ranked second in terms of the highest results, the other feature outcomes were varied. The effect of reducing the features from eight to (five had no significant impact on the results, as the results were very similar, especially when using log energy entropy. EC achieves better results than the other classifiers, whereas the others were acceptable in both cases. Figures S9 and S10 show the ROC curves to determine which classification algorithm is most suitable for both current cases. With high precision and recall for both electrodes above, the classifiers appear to perform well in properly classifying schizophrenia, and results showed that the model can learn effectively.

**Table 9 Two-second epoch size classification performance results of four classifiers.**

| CT | FE | Evaluation metrics | | | | | | | | |
|----|----|-----|-----|-----|-----|-----|----------|-----|-----------------------|--------------------|
|    |    | Acc | Sen | Spe | Pre | NPV | F1-score | MCC | Prediction speed (sec) | Training time (sec) |
| SVM | FFT | 97 | 97 | 97 | 96 | 97 | 96 | 94 | 78,870.56 | 14.18 |
| KNN |  | 94 | 94 | 95 | 93 | 95 | 93 | 88 | 1,109.58 | 55.65 |
| QDA |  | 93 | 90 | 96 | 94 | 92 | 92 | 86 | 133,488.53 | 1.87 |
| EC |  | 96 | 95 | 97 | 96 | 96 | 95 | 92 | 203.4 | 332.7 |
| SVM | ApEn + Bandpass | 77 | 73 | 81 | 79 | 76 | 76 | 54 | 1,7452.77 | 39.71 |
| KNN |  | 71 | 62 | 88 | 91 | 55 | 74 | 48 | 844.01 | 77.51 |
| QDA |  | 71 | 61 | 97 | 98 | 49 | 75 | 52 | 138,362.86 | 1.753 |
| EC |  | 76 | 68 | 89 | 91 | 65 | 77 | 57 | 211.1 | 315.9 |
| SVM | ApEn | 86 | 84 | 88 | 86 | 87 | 85 | 72 | 57,978.78 | 444.40 |
| KNN |  | 85 | 83 | 88 | 85 | 85 | 84 | 70 | 8,527.96 | 9.11 |
| QDA |  | 78 | 69 | 90 | 91 | 66 | 79 | 58 | 319,534.75 | 1.45 |
| EC |  | 85 | 81 | 89 | 87 | 83 | 84 | 70 | 1,033.5 | 52.7 |
| SVM | Shannon entropy | 60 | 78 | 58 | 20 | 95 | 32 | 23 | 1,700.017 | 123.67 |
| KNN |  | 90 | 87 | 92 | 91 | 88 | 89 | 79 | 1,245.70 | 58.06 |
| QDA |  | 69 | 61 | 95 | 97 | 45 | 75 | 49 | 50,695.47 | 3.97 |
| EC |  | 96 | 95 | 97 | 97 | 96 | 96 | 93 | 32,070 | 147.2 |
| SVM | Log energy entropy | 99 | 99 | 99 | 99 | 99 | 99 | 99 | 42,483.20 | 26.64 |
| KNN |  | 99 | 98 | 99 | 99 | 99 | 99 | 98 | 1,096.95 | 57.44 |
| QDA |  | 97 | 94 | 100 | 100 | 95 | 97 | 94 | 133,697.57 | 1.78 |
| EC |  | 99 | 98 | 98 | 99 | 99 | 99 | 98 | 204.5921 | 388.2029 |
| SVM | Kurtosis | 73 | 70 | 76 | 70 | 76 | 70 | 46 | 43,297.83 | 52.54 |
| KNN |  | 67 | 67 | 67 | 52 | 79 | 59 | 33 | 6,695.99 | 11.71 |
| QDA |  | 51 | 48 | 89 | 98 | 13 | 64 | 20 | 162,253.80 | 1.56 |
| EC |  | 74 | 68 | 81 | 80 | 69 | 73 | 49 | 16,782.5 | 423.9 |

**Note:**
Where Classifier Types (CT), Feature Extraction (FE), Accuracy (Acc), Sensitivity (Sen), Specificity (Spe), Precision (Pre).

## Five-second epoch size

In the last scenario, a 5-s epoch size was used, in the first case, the data was used without changing or modifying anything (no augmentation or features reduction). As shown in Table 14, the four classifiers are identical to the two scenarios above and got the highest result using the SVM with the Log Energy Entropy feature. For the highest accuracy here, the sensitivity, specificity, precision, NPV, F1-Score, and MCC were 98%, 99%, 98%, 98%, and 97% for each parameter, respectively. Also, we illustrated the confusion matrix values in Table S11. Thus, we demonstrated that our model yielded high classification performance, and the area under the curve rate of each ROC is higher than 99%, as presented in Fig. S11.

Table S12 shows the confusion matrices for each classifier when using the SNR signal as a dataset for the feature extraction techniques. Each of the outcomes from the SNR approach worked perfectly when the LogEn feature with the band-pass filter was used. Furthermore, Table 15 displays 99% for all classifiers and indicates the measured

**Table 10 Two-second epoch size classification performance results with SNR for four classifiers.**

| CT | FE | Evaluation metrics | | | | | | | | |
|----|----|-----|-----|-----|-----|-----|----------|-----|------------------------|--------------------|
| | | Acc | Sen | Spe | Pre | NPV | F1-score | MCC | Prediction speed (sec) | Training time (sec) |
| SVM | FFT | 96 | 95 | 96 | 95 | 96 | 95 | 91 | 26,290.9 | 90.2 |
| KNN | | 94 | 95 | 93 | 92 | 96 | 93 | 88 | 954.8 | 334.4 |
| QDA | | 94 | 93 | 95 | 95 | 94 | 94 | 89 | 78,916.6 | 7.8 |
| EC | | 96 | 96 | 97 | 96 | 96 | 96 | 93 | 102.5 | 1,194.1 |
| SVM | ApEn | 74 | 73 | 75 | 69 | 79 | 71 | 48 | 5,980.7 | 207.9 |
| KNN | | 73 | 71 | 74 | 67 | 77 | 69 | 45 | 6,348.8 | 284.6 |
| QDA | | 64 | 70 | 62 | 35 | 88 | 47 | 28 | 189,720.1 | 1.1 |
| EC | | 72 | 69 | 74 | 68 | 75 | 69 | 43 | 503.7 | 658.7 |
| SVM | ApEn + Bandpass | 78 | 74 | 81 | 79 | 77 | 76 | 56 | 2,919.6 | 243.5 |
| KNN | | 74 | 67 | 84 | 85 | 65 | 75 | 51 | 1,017.8 | 347.5 |
| QDA | | 70 | 62 | 84 | 87 | 56 | 72 | 45 | 51,116 | 8.3 |
| EC | | 75 | 68 | 83 | 83 | 68 | 75 | 52 | 104.9 | 1,194 |
| SVM | Shannon entropy + Bandpass | 91 | 86 | 96 | 95 | 87 | 90 | 82 | 8,940 | 120.2 |
| KNN | | 96 | 96 | 96 | 96 | 96 | 96 | 92 | 2,208.3 | 543.3 |
| QDA | | 71 | 61 | 96 | 97 | 49 | 75 | 52 | 39,289.4 | 16.8 |
| EC | | 99 | 99 | 99 | 99 | 99 | 99 | 98 | 16,290.4 | 2,479.8 |
| SVM | Log energy entropy + Bandpass | 99 | 99 | 99 | 99 | 99 | 99 | 99 | 44,797 | 41.8 |
| KNN | | 99 | 99 | 99 | 99 | 99 | 99 | 99 | 1,585.7 | 95.5 |
| QDA | | 99 | 99 | 99 | 99 | 99 | 99 | 99 | 39,550.8 | 13.3 |
| EC | | 99 | 99 | 99 | 99 | 99 | 99 | 99 | 63.9 | 1,408 |
| SVM | Kurtosis + Bandpass | 69 | 65 | 71 | 66 | 71 | 65 | 37 | 1,616.7 | 447.4 |
| KNN | | 61 | 62 | 61 | 38 | 80 | 47 | 21 | 836.8 | 5,14.4 |
| QDA | | 58 | 67 | 57 | 14 | 94 | 24 | 15 | 57,521.9 | 7.9 |
| EC | | 66 | 60 | 73 | 72 | 61 | 66 | 34 | 13,041.8 | 1,274.7 |

**Note:**
Where Classifier Types (CT), Feature Extraction (FE), Accuracy (Acc), Sensitivity (Sen), Specificity (Spe), Precision (Pre).

parameters. Also, the training duration plus prediction speed were satisfactory. The ROC curves are displayed in Fig. S12 to help identify the best classification algorithm.

When employing the stretched signals as a dataset for feature extraction approaches, Table S13 displays the confusion matrices for each classifier. Using the LogEn functionality with the band-pass filter allowed all the Stretch approach's results to function flawlessly. In addition, Table 16 shows the measured parameters and shows 99% for all classifiers. In terms of prediction speed, the EC classifier has excellent accuracy. Impressively, the algorithm's stability and dependability were demonstrated effectively with prediction speed. Moreover, Fig. S13 illustrates the ROC obtained stretch method.

The confusion matrices computed for the datasets consisting of reduced features, are displayed in Tables S14 and S15, respectively. Compared to the findings utilizing 19 electrodes, Tables 17 and 18 demonstrate the classifiers' capacity to classify with fewer features since the former's results were superior and close to the results when 19 electrodes were used. Finally, Figs. S14 and S15 show the ROC for the last scenario.

**Table 11 Two-second epoch size classification performance results with stretch for four classifiers.**

| CT | FE | Evaluation metrics | | | | | | | | |
|----|----|-----|-----|-----|-----|-----|----------|-----|--------------------------|----------------------|
| | | Acc | Sen | Spe | Pre | NPV | F1-score | MCC | Prediction speed (sec) | Training time (sec) |
| SVM | FFT | 96 | 95 | 96 | 95 | 96 | 95 | 92 | 25,162.8 | 100.8 |
| KNN | | 94 | 93 | 94 | 93 | 94 | 93 | 88 | 587.3 | 527.9 |
| QDA | | 92 | 90 | 94 | 92 | 91 | 91 | 84 | 60,337.6 | 12.6 |
| ENS | | 95 | 95 | 96 | 95 | 95 | 95 | 91 | 83.3 | 1,445 |
| SVM | ApEn | 95 | 95 | 95 | 95 | 96 | 95 | 91 | 9,231.9 | 179.3 |
| KNN | | 94 | 95 | 94 | 93 | 96 | 94 | 89 | 2,597.8 | 477.8 |
| QDA | | 87 | 86 | 87 | 85 | 88 | 85 | 74 | 403,344.1 | 2.2 |
| ENS | | 95 | 94 | 96 | 95 | 95 | 95 | 90 | 912.2 | 701 |
| SVM | ApEn + Bandpass | 83 | 78 | 88 | 86 | 81 | 82 | 67 | 4,478.3 | 244.4 |
| KNN | | 76 | 68 | 85 | 84 | 69 | 75 | 53 | 637.2 | 684.1 |
| QDA | | 73 | 62 | 96 | 97 | 55 | 75 | 55 | 60,083.6 | 23.3 |
| ENS | | 82 | 73 | 94 | 94 | 74 | 82 | 68 | 79 | 1,641.1 |
| SVM | Shannon entropy + Bandpass | 88 | 82 | 95 | 95 | 82 | 88 | 77 | 7,891.3 | 289.2 |
| KNN | | 94 | 93 | 95 | 94 | 94 | 93 | 88 | 1,793.6 | 792.1 |
| QDA | | 67 | 58 | 96 | 97 | 41 | 72 | 46 | 40,134.3 | 19.2 |
| ENS | | 97 | 97 | 98 | 98 | 97 | 97 | 95 | 21,243.5 | 1,947.5 |
| SVM | Log energy entropy + Bandpass | 99 | 99 | 99 | 99 | 99 | 99 | 99 | 52,458.3 | 76.7 |
| KNN | | 99 | 99 | 99 | 99 | 99 | 99 | 98 | 2,132.5 | 190.5 |
| QDA | | 97 | 95 | 99 | 99 | 96 | 97 | 95 | 36,103.9 | 16 |
| ENS | | 99 | 99 | 99 | 99 | 99 | 99 | 99 | 53.5 | 1,976.8 |
| SVM | Kurtosis + Bandpass | 73 | 69 | 78 | 75 | 72 | 72 | 47 | 2,777 | 554.3 |
| KNN | | 68 | 66 | 69 | 60 | 74 | 63 | 35 | 1,378.5 | 775.1 |
| QDA | | 50 | 47 | 85 | 97 | 12 | 64 | 17 | 38,185.8 | 41.2 |
| ENS | | 75 | 66 | 88 | 90 | 63 | 76 | 54 | 44,475.1 | 1,295.8 |

**Note:**
Where Classifier Types (CT), Feature Extraction (FE), Accuracy (Acc), Sensitivity (Sen), Specificity (Spe), Precision (Pre).

Accordingly, and for more investigation of the classifier's ability and to make sure from the evaluation, we used the error bar to illustrate data variability because when you have small data, the range of errors expands. The variability or uncertainty of the outcome values can be known through error bars, with the standard deviation, standard error, and confidence interval error. How standard deviations describe the data you gathered changes from one measurement to the next. On the other hand, standard errors indicate how much the mean result may vary if the entire experiment were redone. Error bars create lines with bar charts extending from the edge or the center of the displayed data point. The length of an error bar aids in revealing the degree of uncertainty surrounding a data point. While a big error bar would suggest that the values are more dispersed and less dependable, a small error bar indicates that the data are concentrated, indicating that the plotted averaged value is more likely. We measured the error bars for all cases (standard, stretch, SNR, MRMR features reduction) that were used in this study. In Fig. 3, for the first case in which

**Table 12 Two-second epoch size classification performance results with five electrodes for four classifiers.**

| CT | FE | Evaluation metrics | | | | | | | | |
|----|----|-----|-----|-----|-----|-----|----------|-----|------------------------|----------------------|
|    |    | Acc | Sen | Spe | Pre | NPV | F1-score | MCC | Prediction speed (sec) | Training time (sec) |
| SVM | FFT | 91 | 88 | 93 | 90.6 | 91.5 | 89.5 | 81.9 | 11,392.9 | 167.3 |
| KNN |  | 90 | 87.7 | 92 | 90 | 89.9 | 89 | 80 | 2,048.9 | 231 |
| QDA |  | 79.6 | 73.5 | 85.8 | 84 | 75.8 | 78.5 | 59.7 | 164,357 | 455.9 |
| ENS |  | 91.6 | 98 | 57 | 92.6 | 84 | 95 | 65 | 255.9 | 470.8 |
| SVM | ApEn | 83 | 82 | 84 | 80 | 85 | 81.5 | 66 | 11,987 | 60.9 |
| KNN |  | 82.5 | 82.8 | 82 | 84 | 80.9 | 83 | 65 | 43,994 | 123.9 |
| QDA |  | 66 | 58 | 78 | 79.5 | 56 | 67 | 36 | 173,921 | 19.1 |
| ENS |  | 81.5 | 82 | 80 | 80 | 82 | 81 | 63 | 13,999.2 | 161.9 |
| SVM | ApEn + Bandpass | 82.6 | 83.5 | 81.8 | 81 | 84 | 82 | 65 | 12,963.1 | 109.5 |
| KNN |  | 72.9 | 64.8 | 88.9 | 92 | 56 | 76 | 50.9 | 5,106.3 | 166.1 |
| QDA |  | 72.9 | 61.6 | 94.5 | 95.5 | 56 | 74.9 | 54 | 254,110.8 | 2.1 |
| ENS |  | 80.9 | 74.5 | 91 | 92.9 | 69.5 | 82.7 | 64 | 348 | 423.7 |
| SVM | Shannon entropy + Bandpass | 72 | 63 | 89.5 | 91.7 | 56.9 | 74.7 | 50.6 | 26,567.5 | 269 |
| KNN |  | 93.8 | 93 | 94 | 94.7 | 92.9 | 94 | 87.7 | 8,410.3 | 299.1 |
| QDA |  | 58.9 | 61.6 | 53.5 | 72.6 | 41 | 66.6 | 14 | 155,539 | 2 |
| ENS |  | 97 | 98 | 95.9 | 96 | 97.9 | 97 | 94 | 26,718.3 | 569.9 |
| SVM | Log energy entropy + Bandpass | 98.6 | 98 | 99 | 99 | 98 | 98.6 | 97 | 104,827.1 | 98 |
| KNN |  | 98 | 97.9 | 98 | 98 | 98 | 98 | 96 | 7,180.3 | 107.2 |
| QDA |  | 94.8 | 96 | 93 | 93 | 96 | 94.7 | 89.6 | 159,914.8 | 17.8 |
| ENS |  | 98.7 | 99 | 98 | 98 | 99 | 98.5 | 97 | 195.5 | 488.7 |
| SVM | Kurtosis + Bandpass | 70 | 67.6 | 73 | 71.8 | 69 | 69.6 | 41 | 8,970.6 | 149.1 |
| KNN |  | 62 | 59 | 90 | 98 | 20 | 74 | 30 | 624,973 | 209.8 |
| QDA |  | 50.7 | 48 | 61 | 85.5 | 20 | 61.8 | 7 | 162,115 | 2.2 |
| ENS |  | 72 | 62.5 | 83 | 81 | 65.5 | 70.7 | 46 | 15,123 | 631.3 |

**Note:**
Where Classifier Types (CT), Feature Extraction (FE), Accuracy (Acc), Sensitivity (Sen), Specificity (Spe), Precision (Pre).

the data used was standard data, the error bars are relatively small if the window size is 2 s compared to the window size of 1 and 3 s, indicating low variance.

The error bars in Fig. 4, for the second scenario in which the SNR method was used, the error rate is greater than the first case in all cases, and this indicates that when using this technique (SNR), the error rate will be greater.

While the third case is when using a stretch dataset, Fig. 5 shows the three windows used (1, 2, and 5), where the error rate was varied in the window of 1 and 2 s, but the difference is very clear when the window was 5 s in size.

In the last case, when using the MRMR method, the results were close in both cases, especially in one and two window sizes Epoch, counter to the five the difference was clear. Figures 6 and 7 indicate that in the case of five features, the possibility of obtaining good results while reducing the impact and thus enabling greater consistency in outcomes between several windows. In general, the results obtained were promising and indicate that it is possible to classify schizophrenia even in the case of reduced features.

**Table 13 Two-second epoch size classification performance results with eight electrodes for four classifiers.**

| CT | FE | Evaluation metrics | | | | | | | | |
|----|----|-----|-----|-----|-----|-----|----------|-----|-------------------------|--------------------|
| | | Acc | Sen | Spe | Pre | NPV | F1-score | MCC | Prediction speed (sec) | Training time (sec) |
| SVM | FFT | 94.9 | 93.5 | 96 | 95 | 94.7 | 94 | 89.7 | 32,176.5 | 98 |
| KNN | | 93.6 | 93.5 | 93.7 | 94 | 93 | 93.8 | 87 | 1,167.8 | 271.9 |
| QDA | | 87.5 | 83 | 91 | 89.8 | 85.6 | 86 | 75 | 105,423.2 | 1.5 |
| ENS | | 95.8 | 95 | 96.6 | 96.6 | 95 | 95.8 | 91.7 | 110.9 | 920.7 |
| SVM | ApEn | 90 | 92 | 88.7 | 90.9 | 90.8 | 91.7 | 81.5 | 7,690 | 188.1 |
| KNN | | 90.6 | 91.8 | 88.5 | 93 | 86.8 | 92 | 80 | 14,108 | 205 |
| QDA | | 77.7 | 81 | 72 | 81 | 72 | 81 | 53.7 | 112,851 | 16.3 |
| ENS | | 88.8 | 88 | 89 | 89 | 88 | 88.8 | 77.7 | 17,121 | 231.5 |
| SVM | ApEn + Bandpass | 81.6 | 76 | 89 | 90.9 | 72.6 | 82.9 | 64.5 | 131 | 182,345 |
| KNN | | 75.5 | 63 | 90 | 88.7 | 67 | 73.8 | 54.6 | 2,849.8 | 175 |
| QDA | | 74 | 96.6 | 67 | 47 | 98.5 | 63.5 | 54 | 141,822.1 | 2 |
| ENS | | 82.7 | 79 | 84.6 | 74 | 88 | 76.7 | 63 | 130 | 673.8 |
| SVM | Shannon entropy + Bandpass | 82.9 | 74 | 94 | 94 | 73.7 | 83 | 68 | 112.5 | 219,342 |
| KNN | | 96.9 | 96 | 97.6 | 97.6 | 96 | 96.9 | 93.9 | 7,342.2 | 241 |
| QDA | | 72 | 60.5 | 97 | 97.7 | 53.7 | 74.7 | 54 | 96,512.3 | 13 |
| ENS | | 97.6 | 95.9 | 99 | 99 | 96 | 97.6 | 95 | 349 | 1,023.9 |
| SVM | Log energy entropy + Bandpass | 98.7 | 99 | 98 | 98 | 99 | 98.7 | 97.5 | 82,751.2 | 32 |
| KNN | | 98.8 | 98.9 | 98.7 | 99 | 98 | 99 | 97.5 | 6,789.7 | 58.1 |
| QDA | | 97.7 | 97 | 98 | 98.7 | 96 | 98 | 95 | 98,451.8 | 2 |
| ENS | | 98.9 | 99 | 98.9 | 99 | 98 | 99 | 97.8 | 147.4 | 644 |
| SVM | Kurtosis + Bandpass | 70 | 63 | 78.8 | 79.5 | 62 | 70 | 41.9 | 5,999 | 171.7 |
| KNN | | 63.5 | 61 | 65.9 | 64 | 62.8 | 62.7 | 27 | 5,055.7 | 254.3 |
| QDA | | 51.5 | 47 | 89.5 | 97.6 | 15.5 | 63.9 | 22 | 74,523 | 14.9 |
| ENS | | 73.9 | 68 | 80 | 80.5 | 68 | 74 | 48.7 | 15,521.6 | 709.9 |

**Note:**
Where Classifier Types (CT), Feature Extraction (FE), Accuracy (Acc), Sensitivity (Sen), Specificity (Spe), Precision (Pre).

According to the findings obtained from the entropy attribute, the best explanatory electrodes were found to be concentrated in the frontal, central, and parietal areas. The frequency of the occipital entropy was distributed among the theta and beta bands while frontal entropy was observed in beta and gamma bands. Frontal gamma band power was shown to be a discriminator between SZ and healthy groups by *Mitra et al. (2015)*. On the other hand, resting state theta band power was shown to increase in SZ patients (*Iglesias-Tejedor et al., 2022*). The literature findings of the resting state EEG of the SZ patients mostly overlap with the reduced features that we obtained. Thus, it seems as a promising approach to reduce the features to be used in the classification of SZ from healthy group. On the other hand, alpha band entropy of the parietal electrode was one of the main features that discriminates the resting state EEG of SZ group from healthy ones.

From the three scenarios above, it can be observed that the window size significantly influences the accuracy values since when the epoch was 1-s in size, accuracy was at its best

**Table 14 Five-second epoch size classification performance results of four classifiers.**

| CT | FE | Evaluation metrics | | | | | | | | |
|---|---|---|---|---|---|---|---|---|---|---|
| | | Acc | Sen | Spe | Pre | NPV | F1-score | MCC | Prediction speed (sec) | Training time (sec) |
| SVM | FFT | 95 | 95 | 96 | 95 | 96 | 95 | 91 | 62,512.1 | 5.5 |
| KNN | | 92 | 92 | 93 | 91 | 93 | 92 | 85 | 2,861.3 | 10.7 |
| QDA | | 93 | 91 | 95 | 94 | 92 | 92 | 86 | 111,541.6 | 1.8 |
| EC | | 95 | 93 | 96 | 95 | 94 | 94 | 90 | 316.7 | 89.5 |
| SVM | ApEn | 78 | 71 | 91 | 92 | 66 | 80 | 60 | 9,289.1 | 8.5 |
| KNN | | 72 | 63 | 90 | 93 | 52 | 75 | 49 | 2,418.4 | 9 |
| QDA | | 73 | 64 | 95 | 97 | 53 | 77 | 54 | 56,887.9 | 1.5 |
| EC | | 83 | 79 | 86 | 83 | 83 | 81 | 65 | 1,293.9 | 18.6 |
| SVM | ApEn + Band-pass | 85 | 82 | 87 | 84 | 85 | 83 | 69 | 73,877.4 | 31.7 |
| KNN | | 84 | 84 | 83 | 77 | 89 | 80 | 66 | 9,025 | 9.4 |
| QDA | | 77 | 67 | 90 | 90 | 67 | 77 | 57 | 43,113 | 3.9 |
| EC | | 77 | 70 | 89 | 91 | 65 | 79 | 58 | 16,629.5 | 53.9 |
| SVM | Shannon entropy | 85 | 89 | 83 | 77 | 92 | 83 | 70 | 56,088.3 | 138.8 |
| KNN | | 89 | 86 | 93 | 91 | 88 | 89 | 79 | 2,021.8 | 16.1 |
| QDA | | 71 | 62 | 94 | 96 | 51 | 75 | 52 | 48,728.9 | 4.1 |
| EC | | 96 | 95 | 97 | 97 | 96 | 96 | 93 | 16,963.8 | 58.1 |
| SVM | Log energy entropy | 99 | 99 | 99 | 99 | 99 | 99 | 98 | 75,324.9 | 2.7 |
| KNN | | 99 | 98 | 99 | 99 | 98 | 98 | 97 | 3,365.5 | 8.6 |
| QDA | | 98 | 95 | 100 | 100 | 96 | 98 | 95 | 120,410.5 | 1.3 |
| EC | | 98 | 98 | 99 | 99 | 98 | 98 | 97 | 16,881.6 | 59.6 |
| SVM | Kurtosis | 70 | 65 | 80 | 71 | 76 | 68 | 46 | 22,516.1 | 10.4 |
| KNN | | 69 | 63 | 74 | 73 | 65 | 68 | 38 | 10,477.2 | 6.3 |
| QDA | | 54 | 49 | 89 | 97 | 18 | 66 | 25 | 145,563.7 | 1.7 |
| EC | | 69 | 63 | 78 | 79 | 62 | 70 | 41 | 17,675.1 | 64.7 |

**Note:**
Where Classifier Types (CT), Feature Extraction (FE), Accuracy (Acc), Sensitivity (Sen), Specificity (Spe), Precision (Pre).

and fell as the epoch size gradually increased or decreased. The second significant result was the evaluation metrics related to window epoch size that got the best accuracy.

Finally, visual assessment assists physicians in the identification of the signal of the disorder and its location in the brain. The power spectrum density can illustrate the difference between the signals of a patient suffering from schizophrenia compared to the signals of a healthy one. The healthy signals will appear homogeneous and constant, whereas the subject signals exhibit inconsistencies and odd rhythms.

## DISCUSSION

For a long time, it was believed that schizophrenia affected both sexes equally often. However, according to more current evidence, males are more likely than women to suffer from schizophrenia (*González-Rodríguez et al., 2023*). The way that diseases progress depending on a person's gender is becoming more and more recognized in medicine. Men typically experience the early signs of schizophrenia before women do. It is well known

**Table 15 Five-second epoch size classification performance results of four classifiers with SNR augmented method.**

| CT | FE | Evaluation metrics | | | | | | | | |
|---|---|---|---|---|---|---|---|---|---|---|
| | | Acc | Sen | Spe | Pre | NPV | F1-score | MCC | Prediction speed (sec) | Training time (sec) |
| SVM | FFT | 94 | 94 | 94 | 93 | 95 | 93 | 94 | 23,731 | 22.3 |
| KNN | | 92 | 93 | 92 | 89 | 95 | 91 | 92 | 1,892.1 | 100.4 |
| QDA | | 93 | 92 | 94 | 93 | 93 | 92 | 93 | 42,874.9 | 2.1 |
| EC | | 95 | 95 | 95 | 94 | 96 | 95 | 95 | 260.7 | 271.8 |
| SVM | ApEn | 91 | 91 | 91 | 89 | 92 | 90 | 82 | 27,528.2 | 20.6 |
| KNN | | 90 | 90 | 91 | 89 | 92 | 89 | 81 | 8,871.1 | 59.7 |
| QDA | | 77 | 83 | 74 | 63 | 89 | 72 | 56 | 156,474.5 | 0.76 |
| EC | | 92 | 90 | 93 | 92 | 92 | 91 | 84 | 1,746.3 | 96.4 |
| SVM | ApEn + Bandpass | 87 | 85 | 88 | 85 | 88 | 85 | 74 | 22,700.2 | 22.7 |
| KNN | | 83 | 81 | 85 | 82 | 84 | 82 | 67 | 2,428.7 | 48.6 |
| QDA | | 79 | 72 | 87 | 87 | 72 | 79 | 60 | 58,816.2 | 2.5 |
| EC | | 88 | 84 | 92 | 91 | 86 | 88 | 77 | 234 | 243.1 |
| SVM | Shannon entropy + Bandpass | 91 | 86 | 96 | 96 | 87 | 90 | 82 | 22,872.7 | 18.3 |
| KNN | | 97 | 97 | 97 | 96 | 98 | 97 | 95 | 7,718.8 | 56.5 |
| QDA | | 75 | 65 | 95 | 96 | 58 | 78 | 58 | 36,348.4 | 1.7 |
| EC | | 99 | 99 | 99 | 99 | 99 | 99 | 98 | 16,828.4 | 283.8 |
| SVM | Log energy entropy + Bandpass | 99 | 99 | 99 | 99 | 99 | 99 | 99 | 49,353.9 | 13.7 |
| KNN | | 99 | 99 | 99 | 99 | 99 | 99 | 99 | 5,558.3 | 34.2 |
| QDA | | 99 | 99 | 100 | 100 | 99 | 99 | 99 | 50,101.1 | 2.2 |
| EC | | 99 | 99 | 99 | 99 | 99 | 99 | 99 | 272.2 | 185.6 |
| SVM | Kurtosis + Bandpass | 71 | 68 | 74 | 68 | 74 | 68 | 42 | 20,627.9 | 43.2 |
| KNN | | 64 | 60 | 66 | 57 | 69 | 59 | 27 | 1,905.3 | 71.1 |
| QDA | | 59 | 59 | 60 | 35 | 79 | 44 | 17 | 49,366.9 | 2.6 |
| EC | | 72 | 66 | 77 | 76 | 68 | 71 | 44 | 12,206.5 | 343.2 |

Note:
Where Classifier Types (CT), Feature Extraction (FE), Accuracy (Acc), Sensitivity (Sen), Specificity (Spe), Precision (Pre).

that there are gender variations in schizophrenia, which are said to have an impact on several areas, such as symptom profiles, illness progression, and disease start (*Ayesa-Arriola et al., 2020*).

The age range when it occurs most frequently in men is 18 to 25, but for women, it happens around 4 years later (*Gogos et al., 2019*; *Zorkina et al., 2021*). As well, during menopause, women again face a second wave of disease (*Barker & Vigod, 2023*). Schizophrenia progresses more severely in men (*Esposito et al., 2024*). Although gender differences in schizophrenia are usually noted after diagnosis, a recent review indicates that they may exist before clinically detectable symptoms appear, with men showing worse premorbid functioning than women, such as more social disengagement, isolation, and inadequate self-care (*Hoffman et al., 2022*).

Apart from the gender disparities in social and behavioral areas, males are more likely than women to exhibit anomalies in brain morphology when diagnosed with schizophrenia. Gross anatomical examinations, for instance, reveal more severe frontal

**Table 16 Five-second epoch size classification performance results of four classifiers with stretch signals.**

| CT | FE | Evaluation metrics | | | | | | | | |
|---|---|---|---|---|---|---|---|---|---|---|
| | | Acc | Sen | Spe | Pre | NPV | F1-score | MCC | Prediction speed (sec) | Training time (sec) |
| SVM | FFT | 94 | 94 | 95 | 93 | 95 | 94 | 89 | 21,108.7 | 28.4 |
| KNN | | 92 | 94 | 90 | 88 | 95 | 91 | 84 | 2,407.5 | 67.4 |
| QDA | | 92 | 91 | 93 | 92 | 93 | 92 | 85 | 69,198.1 | 2.2 |
| EC | | 94 | 94 | 94 | 92 | 95 | 93 | 88 | 212.1 | 263.9 |
| SVM | ApEn | 94 | 94 | 95 | 94 | 95 | 94 | 89 | 193,844.3 | 282.6 |
| KNN | | 95 | 95 | 95 | 93 | 96 | 94 | 90 | 7,429.2 | 56.2 |
| QDA | | 88 | 87 | 89 | 87 | 89 | 87 | 76 | 234,666.4 | 0.8 |
| EC | | 96 | 95 | 96 | 95 | 96 | 95 | 92 | 992.4 | 107.1 |
| SVM | ApEn + Bandpass | 92 | 89 | 93 | 92 | 91 | 91 | 84 | 21,119.6 | 24.7 |
| KNN | | 89 | 85 | 93 | 92 | 86 | 88 | 79 | 1,420.4 | 81.8 |
| QDA | | 80 | 71 | 94 | 95 | 68 | 81 | 64 | 60,932.5 | 2.5 |
| EC | | 94 | 91 | 97 | 96 | 92 | 93 | 88 | 237.4 | 234.5 |
| SVM | Shannon entropy + Bandpass | 90 | 85 | 96 | 95 | 86 | 90 | 81 | 29,809 | 21.3 |
| KNN | | 96 | 96 | 96 | 95 | 96 | 96 | 92 | 7,885 | 63.8 |
| QDA | | 73 | 63 | 95 | 96 | 53 | 76 | 54 | 79,655.8 | 1.5 |
| EC | | 98 | 97 | 98 | 97 | 98 | 97 | 96 | 27,852.6 | 357.2 |
| SVM | Log energy entropy + Bandpass | 99 | 99 | 99 | 99 | 99 | 99 | 99 | 39,936.6 | 26.3 |
| KNN | | 99 | 99 | 99 | 99 | 99 | 99 | 99 | 5,976.8 | 37.3 |
| QDA | | 99 | 98 | 99 | 99 | 99 | 99 | 98 | 61,461.4 | 2.5 |
| EC | | 99 | 99 | 99 | 99 | 99 | 99 | 99 | 177.3 | 279.8 |
| SVM | Kurtosis + Bandpass | 74 | 70 | 79 | 76 | 73 | 73 | 49 | 19,357.3 | 58.2 |
| KNN | | 67 | 67 | 67 | 55 | 78 | 60 | 34 | 4,651.1 | 77.7 |
| QDA | | 55 | 50 | 83 | 94 | 23 | 65 | 24 | 70,696.9 | 2.1 |
| EC | | 78 | 72 | 86 | 86 | 72 | 78 | 59 | 11,186.7 | 375 |

Note:
Where Classifier Types (CT), Feature Extraction (FE), Accuracy (Acc), Sensitivity (Sen), Specificity (Spe), Precision (Pre).

and temporal lobe atrophy in men and larger ventricular enlargement (*Petric et al., 2024*). Moreover, males have more anomalies in the microstructure of white matter (*Chatterjee et al., 2020*).

SZ comprises disorders with somewhat overlapping brain correlates and phenomenology (*Nagy et al., 2023*). Moreover, SZ appears to result from intricate interactions between endogenous and exogenous variables that influence neurodevelopment. Schizophrenia and other psychotic diseases are characterized by five essential characteristics. Delusions, hallucinations, wildly chaotic or aberrant motor action, disorganized thought processes (inferred from speech) (*Baandrup, 2020*). Treating SCZ at its earliest stage may assist in slowing down the disease's course.

Many articles have been published aimed at classifying schizophrenia diseases using different types of techniques such as MRI (*Tyagi, Singh & Gore, 2022*), genetics (DNA) (*Yu et al., 2022*), eye tracking (*Lim, Mountstephens & Teo, 2022*), facial features (*Rahman et al., 2021*, *Wang & Wang, 2021*), tracking handwriting (*Rashid et al., 2020*), speech (*Lopez-*

**Table 17 Five-second epoch size classification performance results with five electrodes for four classifiers.**

| CT | FE | Evaluation metrics | | | | | | | | |
|----|----|-----|-----|-----|-----|-----|---------|-----|------------------------|---------------------|
| | | Acc | Sen | Spe | Pre | NPV | F1-score | MCC | Prediction speed (sec) | Training time (sec) |
| SVM | FFT | 88.6 | 84.8 | 92 | 91 | 86 | 87.9 | 77 | 36,398.2 | 19.5 |
| KNN | | 90 | 90 | 91 | 91 | 90.6 | 90.7 | 81.6 | 8,167.8 | 28.1 |
| QDA | | 82 | 82 | 81.9 | 85.7 | 78 | 84 | 64 | 101,433 | 1.4 |
| ENS | | 90.8 | 87 | 92.8 | 87.9 | 92.5 | 87.6 | 80 | 505.7 | 98.2 |
| SVM | ApEn | 85.8 | 85.5 | 86 | 89.9 | 80.6 | 87.6 | 71 | 36,288 | 25.4 |
| KNN | | 84.8 | 83.7 | 85.7 | 82 | 86.8 | 83 | 69 | 42,568 | 28.1 |
| QDA | | 65 | 67.7 | 64 | 46.9 | 80.8 | 55 | 29 | 89,231.5 | 12.9 |
| ENS | | 80.6 | 72 | 86 | 77 | 82.5 | 74.8 | 59 | 12,184.2 | 44 |
| SVM | ApEn + Bandpass | 87 | 87.9 | 85.8 | 91 | 79.9 | 89.8 | 72.7 | 69,544.4 | 21.9 |
| KNN | | 82.9 | 81.5 | 84 | 82 | 83.6 | 81.8 | 65.8 | 7,745.6 | 29.9 |
| QDA | | 70.9 | 69 | 83 | 96 | 29.9 | 80 | 37 | 64,465 | 1.5 |
| ENS | | 87.8 | 83.7 | 92 | 91 | 84.7 | 87 | 75.9 | 598.8 | 92 |
| SVM | Shannon entropy + Bandpass | 76.9 | 71.8 | 89 | 94 | 57 | 81 | 55.9 | 26,398 | 61.9 |
| KNN | | 95 | 93 | 97 | 97 | 93 | 95 | 90 | 14,132.5 | 68.3 |
| QDA | | 72 | 68.7 | 79 | 86.9 | 55.8 | 76.7 | 45 | 81,321.1 | 2.1 |
| ENS | | 97.8 | 98 | 97 | 98 | 96.5 | 98 | 95 | 15,212.6 | 139.3 |
| SVM | Log energy entropy + Bandpass | 98.6 | 98 | 99 | 99 | 98 | 98.6 | 97 | 91,126.7 | 9.1 |
| KNN | | 97.8 | 96 | 99 | 99 | 96 | 97.7 | 95.6 | 15,123 | 12.9 |
| QDA | | 91.7 | 86 | 98 | 98 | 85 | 91.9 | 84 | 183,989 | 1.1 |
| ENS | | 98.9 | 99 | 98.9 | 98.8 | 99 | 98.9 | 97.9 | 456.5 | 92.9 |
| SVM | Kurtosis + Bandpass | 71 | 58.6 | 83 | 76 | 68.6 | 66 | 43 | 39,912.3 | 430 |
| KNN | | 65 | 59.9 | 71 | 70.9 | 60 | 64.9 | 31 | 9,169.5 | 50.8 |
| QDA | | 57.9 | 55 | 67.7 | 85 | 30 | 66.9 | 19 | 83,304 | 2.2 |
| ENS | | 73 | 66.6 | 80 | 81 | 66 | 73 | 47.5 | 11,659.1 | 166.9 |

**Note:**
Where Classifier Types (CT), Feature Extraction (FE), Accuracy (Acc), Sensitivity (Sen), Specificity (Spe), Precision (Pre).

*Bernal et al., 2022*), and schizophrenia EEG signal (*Krishnan et al., 2020*). The last one is the technique used to classify schizophrenia. Since EEG signals are complicated nonlinear dynamic signals, it might be difficult to isolate their constituent parts precisely. In our study, we confirmed the concordance of brain states within different diagnostic features and across three different window sizes, which is a crucial step in linking brain states and capturing the differences inside functional brain activities.

This section summarizes and discusses various work scenarios that have been done and compares article results for classifying schizophrenic patients and healthy controls using the same dataset and different datasets. Also, this part compares state-of-the-art automated identification methods for SZ EEG data. Table 19 shows the most researchers use a single dataset to classify EEG signals with SZ, as well as present a comparison of article results classifying schizophrenic patients and healthy controls using different datasets using different machine-learning techniques for EEG signals. Selection feature extraction playing an essential role in improving classification accuracy, we proposed a

**Table 18 Five-second epoch size classification performance results with eight electrodes for four classifiers.**

| CT | FE | Evaluation metrics | | | | | | | | |
|---|---|---|---|---|---|---|---|---|---|---|
| | | Acc | Sen | Spe | Pre | NPV | F1-score | MCC | Prediction speed (sec) | Training time (sec) |
| SVM | FFT | 92.5 | 94 | 90.8 | 91 | 93.7 | 92.7 | 85 | 26,322.1 | 17 |
| KNN | | 93 | 92.8 | 93 | 91.9 | 93.9 | 92 | 85.9 | 5,121.8 | 27 |
| QDA | | 88.7 | 90.8 | 86.7 | 86.5 | 90.9 | 88.6 | 77.5 | 52,143.1 | 2.9 |
| ENS | | 94.8 | 94.6 | 95 | 92.6 | 96 | 93.6 | 89 | 404 | 129 |
| SVM | ApEn | 93.5 | 93.5 | 93.5 | 91.6 | 95 | 92.5 | 86.8 | 42,561.7 | 28 |
| KNN | | 93 | 94 | 92 | 92.8 | 93.8 | 93.6 | 86.6 | 28,679 | 19 |
| QDA | | 78.8 | 86 | 71.7 | 74.6 | 84.5 | 80 | 58.5 | 69,534.8 | 14.9 |
| ENS | | 90.8 | 91 | 90.6 | 94.5 | 85 | 92.7 | 80.5 | 6,980.3 | 35.5 |
| SVM | ApEn + Bandpass | 90 | 89 | 91 | 90.8 | 89.5 | 89.9 | 80 | 43,616.2 | 13.3 |
| KNN | | 87 | 83 | 91 | 91 | 83.6 | 87 | 74.6 | 5,761.7 | 28.2 |
| QDA | | 75.8 | 63 | 95 | 95 | 62.6 | 76 | 58 | 15,432.8 | 1 |
| ENS | | 92.8 | 90 | 95.6 | 96 | 89 | 93 | 85.7 | 533.1 | 110.6 |
| SVM | Shannon entropy + Bandpass | 82.8 | 74 | 93.6 | 93.5 | 74.5 | 82.7 | 67.9 | 45,943.2 | 27.3 |
| KNN | | 96.9 | 95 | 98 | 97.9 | 96 | 96.5 | 93.7 | 13,144.9 | 48.2 |
| QDA | | 69 | 60 | 94 | 96.9 | 45 | 74 | 47.8 | 119,905 | 0.7 |
| ENS | | 97 | 97.6 | 97 | 96 | 98 | 97 | 94.6 | 26,179.5 | 138 |
| SVM | Log energy entropy + Bandpass | 98.5 | 97.6 | 99 | 99 | 98 | 98 | 97 | 105,432 | 4 |
| KNN | | 98.9 | 98.6 | 99 | 99 | 98.6 | 98.9 | 97.9 | 137,432.5 | 5 |
| QDA | | 98.9 | 99 | 98.5 | 98.8 | 99 | 99 | 97.9 | 90,543.9 | 1 |
| ENS | | 98.8 | 98 | 99 | 99 | 98 | 98.7 | 97.7 | 421 | 94 |
| SVM | Kurtosis + Bandpass | 73.7 | 72 | 75.8 | 80.6 | 66 | 76 | 47 | 19,697 | 91.9 |
| KNN | | 69 | 64.6 | 72.9 | 66.8 | 70.9 | 65.7 | 37.7 | 8,813.4 | 40 |
| QDA | | 55 | 49 | 76 | 87.7 | 30 | 62.9 | 21 | 74,699.9 | 1 |
| ENS | | 77 | 65 | 87.5 | 82 | 74 | 72.6 | 54 | 13,087 | 159 |

**Note:**

Where Classifier Types (CT), Feature Extraction (FE), Accuracy (Acc), Sensitivity (Sen), Specificity (Spe), Precision (Pre).

method that combined the most important feature methods with a bandpass filter. That's why our results outperformed other researchers and they could not get higher results even when using a bandpass filter. Using a bandpass filter in combination with various preprocessing approaches, the methods by *Kim, Lee & Lee (2020)*, *Azizi, Hier & Wunsch (2021)*, *Kim et al. (2021)*, *Keihani et al. (2022)*, fall short of achieving the remarkable accuracy seen by *Najafzadeh et al. (2021)*, *Aydemir et al. (2022)*, *Agarwal & Singhal (2023)*.

Next, Table 20 displays the outcomes of the articles that classified schizophrenia disorder using the same dataset. Various factors affected the final results of classification, such as the number of electrodes, size of the sample rate, number of subjects, and many others. Thus, the results we observed in this research related to the features and characteristics that were used and to the normalization techniques that were applied to the extracted features. Based on our results, our proposed technique can classify healthy and schizophrenic subjects and show clear differences in their brain activity based on power

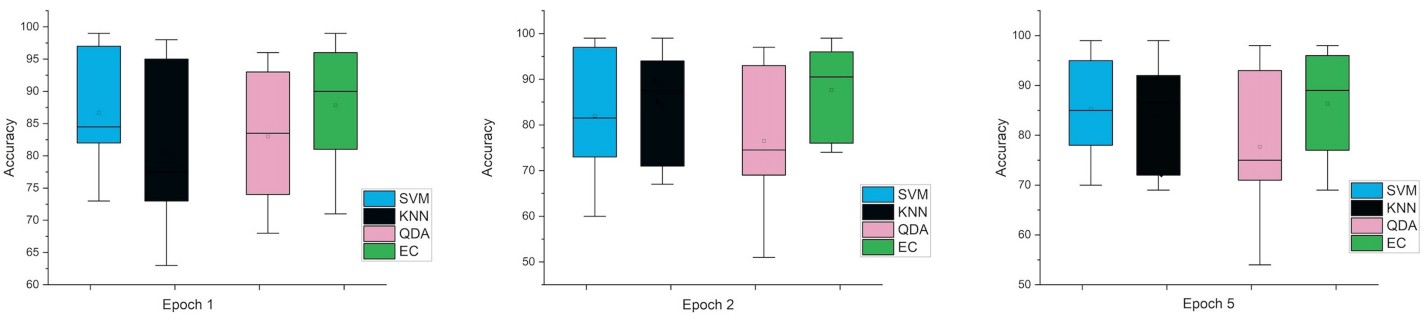

**Figure 3** The mean error bars for the accuracy of the classification SZ using three Epoch window sizes with four classifiers based on standard dataset.

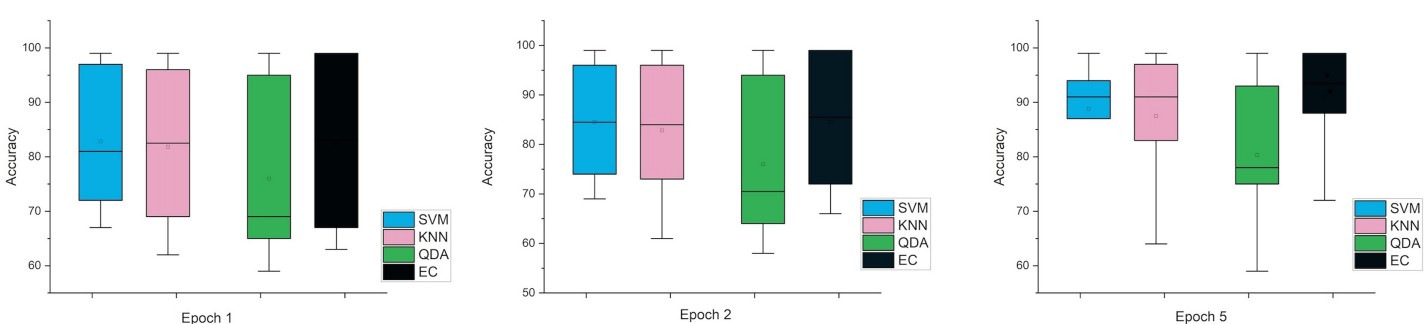

**Figure 4** The mean error bars for the accuracy of the classification SZ using three Epoch window sizes with four classifiers based on the SNR dataset.

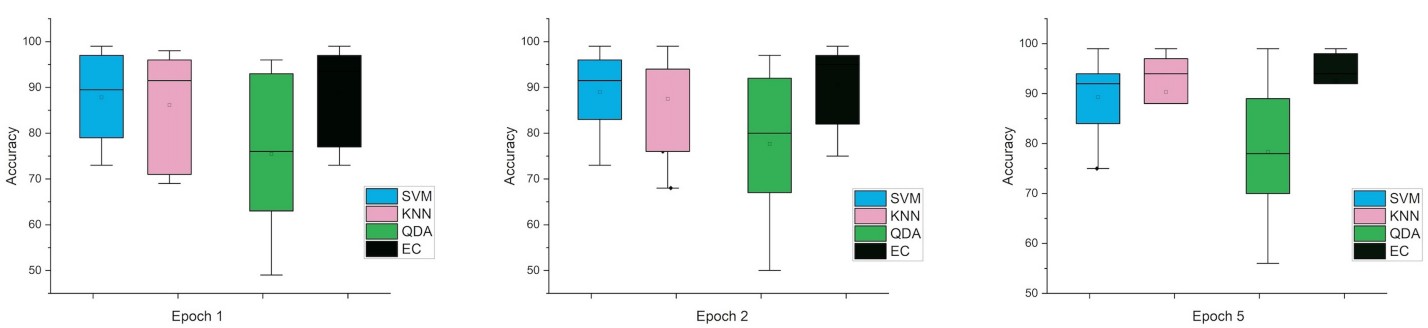

**Figure 5** The mean error bars for the accuracy of the classification SZ using three Epoch window sizes with four classifiers based on the stretch dataset.

spectrum and topography images. Thus, our method showed outperformance when compared to these articles.

For time-series datasets, the augmentation includes techniques such as generating data, shuffling features, time warping, and adding white Gaussian noise to signal (SNR) to the original dataset. Among these, we used two different techniques: the SNR and the stretch

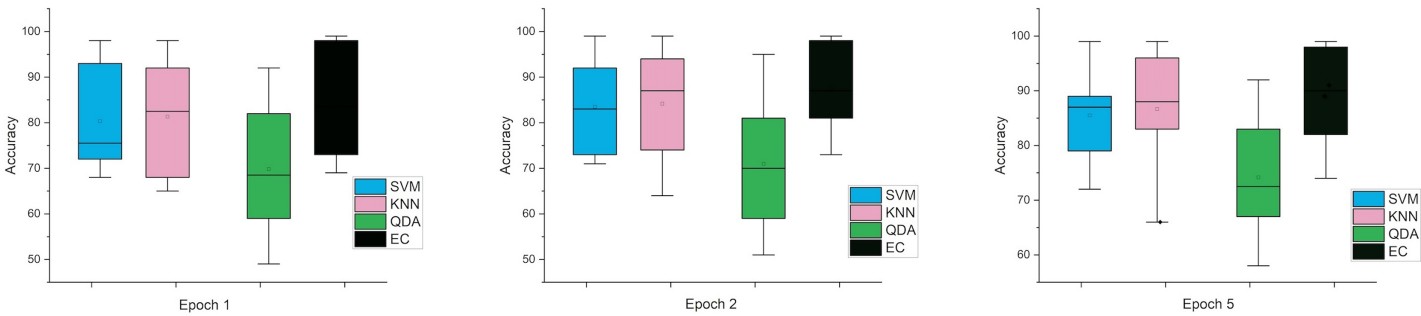

**Figure 6** The mean error bars for the accuracy of the classification SZ using three Epoch window sizes with four classifiers based on the five electrodes dataset.

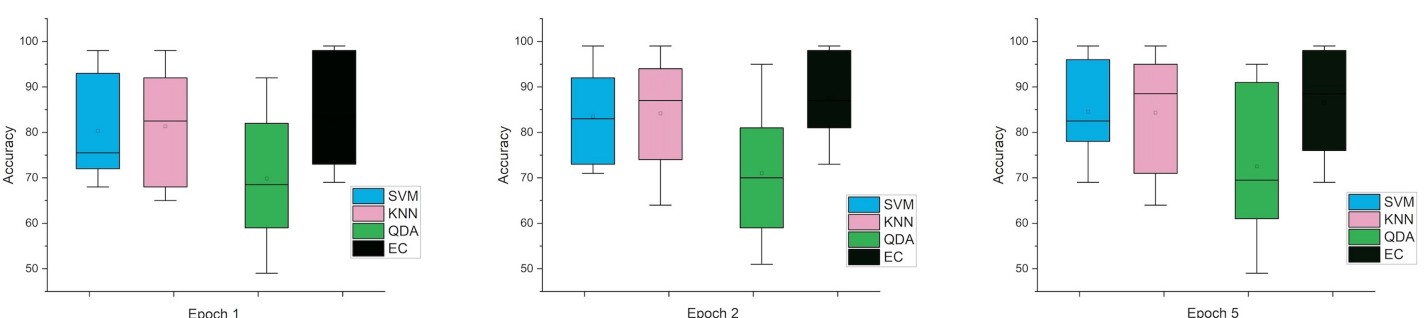

**Figure 7** The mean error bars for the accuracy of the classification SZ using three epoch window sizes with four classifiers based on the eight electrodes dataset.

method to increase sample size which finally led to improving training and testing classifiers. Then, a bandpass filter was applied to the augmented data and the sub-bands were used as input to feature extraction to obtain the hidden information between the signals to prepare it as input to the classifiers.

For the augmentation approach, first, according to which was calculated using the LogEn method with the four classifiers, and the results for the performance showed that LogEn outperformed the other feature indices with higher computational efficiency. Second, in terms of accuracy, the results were obtained by using the stretch method and the Log Energy Entropy features with all classifiers, ranged between 96 to 99, which is the highest results obtained. FFT was second in terms of accuracy, and SVM obtained the highest accuracy of 97%, while the lowest accuracy obtained by using QDA was 92%. For the rest classifiers, the results ranged from 92% to 96%, which is relatively close to the first method. The rest of the classifiers, the results obtained in all criteria were acceptable. From the affirmative results, we concluded the window size that obtained the highest results was 1 s.

The final implementation was to use f the MRMR features selection methods by selection (the eight and five best features). The highest results were also for the log energy

**Table 19 Comparison of the sensitivity (Sen)/specificity (Spec)/ accuracy (Acc) values as a result of our proposed method with articles using various datasets.**

| Study | Year | Electrodes no. | Classes | Dataset | | Applied algorithms | Acc% | Spec | Sen |
|---|---|---|---|---|---|---|---|---|---|
| | | | | Patients | Control | | | | |
| de Miras et al. (2023) | 2023 | 31 | 2 | 9♂, 20♀ | 13♂, 7♀ | SVM | 89 | 90 | 63 |
| Zhang (2019) | 2019 | 64 | 2 | 49 | 32 | RF | 81 | NA | NA |
| Li et al. (2019) | 2019 | 16 | 2 | 37♂, 10♀ | 14♂, 11♀ | SVM | 90.48 | 91.30 | 89.47 |
| Winterburn et al. (2019) | 2019 | NA | 2 | 31♂, 19♀ | 32♂, 18♀ | Non-linear SVM | 73.5 | 56.3 | 62.5 |
| | 2020 | 8 | 2 | 48 | 24 | RF | 68 | NA | NA |
| | 2022 | NA | 2 | 158 | 76 | Ensemble | 87 | 65 | 98 |
| | 2022 | 64 | 2 | 41♂, 8♀ | 67♂, 14♀ | HDSS | 92.93 | 91.06 | 97.15 |
| Baradits, Bitter & Czobor (2020) | 2020 | 256 | 2 | 33♂, 37♀ | 47♂, 28♀ | SVM | 82.7 | 81.43 | 82.67 |
| | 2023 | 32 | 2 | 310 | 205 | XGB | 94 | NA | NA |
| Zandbagleh et al. (2022) | 2022 | 64 | 2 | 13 | 11 | SVM | 89.21 | 90.3 | 88.2 |
| Ciprian et al. (2021) | 2021 | 20 | 2 | 37♂, 25♀ | 38♂, 32♀ | KNN | 96.92 | 98.57 | 95 |
| Phang et al. (2019b) | 2019 | 16 | 2 | 45 | 39 | MDC-CNN | 93.06 | 93.33 | 93.33 |
| Sun et al. (2021) | 2021 | 64 | 2 | 36♂, 18♀ | 31♂, 24♀ | Hybrid DNN | 99.22 | NA | NA |
| Guo et al. (2021) | 2021 | 64 | 2 | 49 | 32 | CNN | 92 | NA | NA |
| Bretones et al. (2023) | 2023 | 32 | 2 | 215♂, 97♀ | 176♂, 144♀ | RBF | 93.40 | NA | NA |
| Present | 2024 | 19 | 2 | 7♂, 7♀ | 7♂, 7♀ | SVM | 99.9 | 99.9 | 99.9 |

**Table 20 Comparison of the sensitivity (Sen)/specificity (Spec)/ accuracy (Acc) values as a result of our proposed method with articles using the same datasets.**

| Study | Year | Electrodes no. | Classes | Dataset | | Applied algorithms | Acc% | Spec% | Sen% |
|---|---|---|---|---|---|---|---|---|---|
| | | | | Patients | Control | | | | |
| Akbari et al. (2021) | 2021 | 19 | 2 | 7♂, 7♀ | 7♂, 7♀ | KNN | 94.8 | 95.2 | 94.3 |
| Jahmunah et al. (2019) | 2019 | | | | | SVM | 92.91 | NA | NA |
| Prabhakar, Rajaguru & Lee (2020) | 2020 | | | | | AdaBoost | 98.77 | NA | NA |
| Shalbaf, Bagherzadeh & Maghsoudi (2020) | 2020 | | | | | SVM | 98.60 | 96.92 | 99.65 |
| Oh et al. (2019) | 2019 | | | | | SoftMax | 98.07 | 98.17 | 97.32 |
| Shoeibi et al. (2021) | 2021 | | | | | Sigmoid | 99.25 | NA | NA |
| Aslan & Akin (2020) | 2022 | | | | | SoftMax | 99.5 | NA | NA |
| Das & Pachori (2021) | 2021 | | | | | SVM | 98.9 | 98.8 | 99.1 |
| Krishnan et al. (2020) | 2020 | | | | | SVM | 93 | 93.33 | 98 |
| Buettner et al. (2020) | 2020 | | | | | Random forest | 96.77 | NA | NA |
| Chandran, Sreekumar & Subha (2021) | 2021 | | | | | SVM | 99 | NA | 99 |
| Racz et al. (2020) | 2020 | | | | | Random forest | 89.29 | NA | NA |
| Sharma & Acharya (2021) | 2021 | | | | | KNN | 97.20 | 98.06 | 96.49 |
| Bagherzadeh, Shahabi & Shalbaf (2022) | 2022 | | | | | Boosted Trees | 96.26 | 97.02 | 95.48 |
| Hassan, Hussain & Qaisar (2023) | 2023 | | | | | LR | 98 | 97 | 99 |
| Gosala et al. (2023) | 2023 | | | | | DT | 97.98 | 97.72 | 98.2 |
| Kumar et al. (2023) | 2023 | | | | | AdaBoost | 99.36 | 100 | 98.8 |
| Agarwal & Singhal (2023) | 2023 | | | | | BT | 96.12 | 96.99 | 95.20 |
| Ours | 2024 | | | | | SVM | 99.9 | 99.9 | 99.9 |

entropy in both cases (five and eight features). The results ranged from 97.9 to 98.9 in all epoch sizes for the classifiers (SVM, KNN, EC) in the case of eight-features in all window sizes except the classifier QDA, the accuracy results obtained ranged from 94.8% to 98.9%. Next, the five-features results were less when using eight-features, as observed in window length 1 s. The results were 1% smaller in the classifiers (SVM, KNN). For the rest of the classifiers, the results were the same. It is noted that the results obtained using FFT in both cases (five- features and eight- features) were relatively close, as the difference was slight at 3% in Window lengths 1 and 2 s, while when using window length 5 s the ratio difference was 4%. For the rest of the features and classifiers used, the results are lower than those mentioned above. It is very noticeable that the best result was obtained by using window 5 s when dealing with the eight features, and this indicates the possibility of reducing the features to a certain percentage without affecting the accuracy considerably.

Noteworthy, in a recent study, by *Haider et al. (2024)*, applied the same dataset used with the electrode reduction method (6-electrode) which in turn led to features reduction also, they used the penalized sequential dictionary learning (PSDL) technique to classify the SZ subjects and achieved 89.12%. Interestingly, our results indicated that our MRMR features selection methods (five and eight features) have captured important information that is captured by the three used epochs, yielding higher overall accuracy rates. Eventually, the abovementioned results, the LogEn features calculated with SVM ns EC classifiers recorded as the best method for classifying schizophrenic patients. This indicates that the proposed method was better according to all the criteria by which it was measured.

The proposed SVM in high-dimensional spaces is very effective, and for the function of decision, it uses a subset of training points, which makes it efficient in memory. In contrast, the LogEn technique gives the most accurate characteristics for EEG classification with an absolute error as low as 0.01. LogEn values are signal features that characterize the degree of EEG complexity. Moreover, LogEn determines the optimum number of hidden neurons in hidden layers. LogEn allows us to identify seizure activity from seizure-free epileptic activity with excellent accuracy, even with few characteristics. Therefore, combining SVM and LogEn could be a reason for significant performance improvement. The results show that classifiers alone are unsuccessful at recognizing SZ. However, integrating SVM and log energy entropy features dramatically enhanced SZ detection performance. As shown in Table 11, our proposed architecture has maximum classification accuracy when LogEn values are used as features.

Our method can support the clinical auxiliary diagnosis by quick diagnosis: by examining EEG rhythms, which could point to abnormalities connected to the disorder, such methods may aid in confirming the diagnosis of schizophrenia with higher accuracy. A Scientific Study: gaining insight into the neurological basis of schizophrenia aids current scientific endeavors. Physicians may discover more about the processes behind the illness and recognize brainwaves. Due to the high accuracy achieved, it is possible to integrate this method into a decision support system that will have a minimum error rate.

In this study, some limitations are listed below.

1) The data set is open access instead of a collected private dataset or one gained from cohort studies. Even though the number of subjects was low, we used the epoching approach that enabled us to increase the size of the input data for machine learning.

2) The dataset in this article aimed to identify the condition rather than to assess its degree of severity. Thus, it does not provide the stage of schizophrenia (prodromal, active, and residual).

## CONCLUSION

A person's life and behavior are affected according to the changing electrical activity of the brain, which the EEG can observe. It may be claimed that a healthy brain functions more actively than a brain affected by schizophrenia.

In this study, we proposed a method to classify schizophrenia based on using an EEG signal dataset that contains 28 subjects: 14 suffering from schizophrenia and 14 healthy controls. Due to the variations of the EEG signal, we applied a band-pass filler to decompose the EEG signal into five sub-bands. Next, we implemented six feature extraction methodologies, we applied the first four (ApEn, LogEn, ShnEn, and kurtosis) to the band-pass filter, and then we used FFT and ApEn again without band-pass filters to compare the impact of the filter on the results.

Normalization was used for all features deduced to be on the same scale using the L2 normalization technique. Consequently, we fed the features to the SVM, EC, KNN, and QDA classifiers. We implemented these classifiers using three window sizes (1, 2, and 5 s). Based on the obtained results, we can see a significant variation in the features of EEG signals between healthy subjects and schizophrenia patients, as presented in the tables above. Since, there are many factors affect why the LogEn achieved the most beneficial feature results, which are: 1) Dimensional reduction: by applying the energy logarithm of the EEG signal and then computing the Entropy, this reduces the features space, which is reflected in subsequent efficient computation. 2) artifact elimination: the logarithm aids in eliminating the noise in the signals by emphasizing the magnitude of EEG signals. 3) Sensitivity of the features: it can sense the distribution frequency bands, which is very valuable with EEG signals, due to using a bandpass filter to divide the signal into sub-bands (delta, theta, alpha, beta, and gamma). 4) Non-linear Dynamics: normally, EEG signals are non-linear and complex; thus, the LogEn deals with the signals dynamically due to its ability to capture features from signals with complexity and irregularity. Furthermore, the LogEn strength is the ability of the probability distribution, which calculates the power spectral density. Then the calculated values are normalized to the total signal energy, and the captured values are used as probabilities. In our study, LogEn allows us to identify SZ with excellent accuracy, even with few characteristics. The LogEn technique gives the most accurate characteristics for EEG classification with a minimum error as low as 0.01. From all classifiers, integrating SVM and EC with LogEn was the most effective with high-dimensional spaces, and for the function of decision, which makes it efficient in memory. Thus, the highest results were achieved using features extracted by LogEn with SVM and EC classifiers.

Based on the results obtained in this study, we conclude:

1) Using these feature extraction methods with this type of signal can reach high results.
2) Window epoch sizes could enhance classification precision.
3) The outcomes fluctuate depending on the window size. As a result, the signal decomposition window-size epoch is important for identifying tiny changes in the EEG recording.

For future work, we will apply wavelet transform to convert the signal to time-frequency images and then apply these images to deep convolutional neural networks.

### Funding
This work was supported by FCT-Fundação para a Ciência e Tecnologia, I.P. by project reference UIDB/50008/2020, and DOI identifier https://doi.org/10.54499/UIDB/50008/2020. This work was also funded by FCT/MEC through national funds and co-funded by the FEDER-PT2020 partnership agreement under the project UIDB/00308/2020 (DOI 10.54499/UIDB/00308/2020). The funders had no role in study design, data collection and analysis, decision to publish, or preparation of the manuscript.

### Grant Disclosures
The following grant information was disclosed by the authors:
FCT-Fundação para a Ciência e Tecnologia: UIDB/50008/2020.
FCT/MEC.
FEDER-PT2020 Partnership Agreement: UIDB/00308/2020.

### Competing Interests
Ivan Miguel Pires and Paulo Jorge Coelho are Academic Editors for PeerJ Computer Science.

### Author Contributions
- Athar Alazzawı conceived and designed the experiments, performed the experiments, analyzed the data, performed the computation work, prepared figures and/or tables, authored or reviewed drafts of the article, and approved the final draft.
- Saif Aljumaili conceived and designed the experiments, performed the experiments, analyzed the data, performed the computation work, prepared figures and/or tables, authored or reviewed drafts of the article, and approved the final draft.
- Adil Deniz Duru conceived and designed the experiments, performed the experiments, analyzed the data, performed the computation work, prepared figures and/or tables, authored or reviewed drafts of the article, and approved the final draft.
- Osman Nuri Uçan conceived and designed the experiments, performed the experiments, analyzed the data, performed the computation work, prepared figures and/or tables, authored or reviewed drafts of the article, and approved the final draft.

- Oğuz Bayat conceived and designed the experiments, performed the experiments, analyzed the data, performed the computation work, prepared figures and/or tables, authored or reviewed drafts of the article, and approved the final draft.
- Paulo Jorge Coelho conceived and designed the experiments, performed the experiments, analyzed the data, performed the computation work, prepared figures and/or tables, authored or reviewed drafts of the article, and approved the final draft.
- Ivan Miguel Pires conceived and designed the experiments, performed the experiments, analyzed the data, performed the computation work, prepared figures and/or tables, authored or reviewed drafts of the article, and approved the final draft.

### Data Availability

The EEG in schizophrenia dataset is available at: Olejarczyk, Elzbieta; Jernajczyk, Wojciech, 2017, "EEG in schizophrenia", https://doi.org/10.18150/repod.0107441, RepOD, V1.

The code is available at Zenodo: Alazzawi, A., Aljumaili, S., Duru, A. D., Uçan, O. N., Bayat, O., Coelho, P. J., & Pires, I. M. (2024). Schizophrenia diagnosis based on diverse epoch size using EEG signal on resting-state. Zenodo. https://doi.org/10.5281/zenodo.11075184.

### Supplemental Information

Supplemental information for this article can be found online at http://dx.doi.org/10.7717/peerj-cs.2170#supplemental-information.

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
