# Peer review of "Schizophrenia diagnosis based on diverse epoch size resting-state EEG using machine learning"

_PeerJ Computer Science, doi:10.7717/peerj-cs.2170_

## Round 0.1 · original submission · Major Revisions

Dear Authors,
Your paper has been revised. Given the reviewers' considerations, it needs major revisions before being considered for publication in PEERJ Computer Science.

More precisely, the following points must be faced and clarified:

1) The dataset presently utilized is relatively tiny. The authors must consider data augmentation techniques to enhance the exploited datasets to obtain more reliable results.

2) The authors' selection of the time window is somewhat arbitrary. To strengthen the study, it is necessary to provide the rationale for its choice. Furthermore, the strategy of segmenting data into four frequency bands must be explained.

3) The inclusion of a confusion matrix may be optional. It should be presented in a more streamlined and appropriate format if deemed necessary.

4) When reviewing the relevant literature, it seems that the diagnostic accuracy of most existing studies is above 80%, which is already considered an excellent model. Therefore, the authors should highlight the innovation of their approach and emphasize the necessity of conducting this study by addressing the shortcomings of current research. Furthermore, they have to cite references from the past five years because some of the references in the paper are relatively old.

**Language Note:** PeerJ staff have identified that the English language needs to be improved. When you prepare your next revision, please either (i) have a colleague who is proficient in English and familiar with the subject matter review your manuscript, or (ii) contact a professional editing service to review your manuscript. PeerJ can provide language editing services - you can contact us at [email protected] for pricing (be sure to provide your manuscript number and title). – PeerJ Staff

Reviewer 1 ·

Basic reporting

The main research question of the study is how to use resting-state electroencephalogram (EEG) signals with various window size epochs for machine learning techniques to diagnose schizophrenia. The author defined this research question to explore a new approach for diagnosing schizophrenia.

Based on the provided information, the author has described enough details in the experimental design to allow others to replicate the experiment, including defining the research question, methods, and ethical standards.

The author provided the original data and statistical analysis to support the credibility of the research results, including comparisons with different datasets and evaluation metrics.

To enhance the paper, the author should delve into several aspects for further research:

1. **Data Augmentation**: The dataset presently utilized is relatively small. It would be beneficial to incorporate multicenter data to assess if the methodology is still robust or not. Given that other centers may employ different electrode configurations, selecting a consistent set of 19 electrodes for comparison and testing could prove advantageous. In deep learning, data augmentation techniques are widely used to enhance datasets by generating more training data, thereby improving the performance and generalization ability of models. These techniques allow for the artificial expansion of the dataset through various transformations of existing data, without directly increasing the volume of original data. You can consider using such techniques.

2. **Electrode Reduction**: Exploring the possibility of utilizing fewer electrodes, such as eight or fewer, could significantly increase the method's ease of use and practical applicability.

3. **Method and Algorithm Explainability**: Enhancing the explainability of the method and algorithm is paramount. Clarification on how such high accuracy is achieved would be invaluable for clinicians. Identifying diagnostically valuable electrodes or frequency bands would add considerable depth to the findings.

4. **Time Window Optimization**: The selection of the time window seems to be somewhat arbitrary. Identifying an optimal, perhaps shorter, time window and providing a rationale for its effectiveness in diagnosis would strengthen the study.

5. **Advanced Methodological Approaches**: The exploration of more sophisticated methods, such as XGBoost or ensemble learning, could potentially elevate the research outcomes.

6. **Confusion Matrix Presentation**: The inclusion of a confusion matrix may not be essential. If deemed necessary, it should be presented in a more streamlined and appropriate format.

7. **Frequency Band Analysis**: The strategy of segmenting data into four frequency bands may not be the most efficacious approach. Direct calculation of various metrics (as features) of spectral spectrums might offer more insightful results and remarkable performance.

Experimental design

no comments.

Validity of the findings

no comments.

Additional comments

The author said
Schizophrenia is a severe mental disorder that impairs a person's mental, social, and
emotional faculties gradually. Detection in the early stages with an accurate diagnosis is
crucial to remedying the patients.

but are you studying detection in EARLY stages?
You do research to classify the patient from controls, not to recognize the persons in their early stages.

Reviewer 2 ·

Basic reporting

no comment.

Experimental design

no comment.

Validity of the findings

no comment.

Additional comments

This research utilizes three diverse machine learning algorithms to construct a diagnostic model for schizophrenia, capitalizing on five key indicators derived from EEG data. The integration of different classification methods and feature sets achieves diagnostic accuracies surpassing 80% for this condition. However, before this work can be deemed suitable for publication, certain critical issues require resolution. If these concerns are adequately addressed, it is the reviewer's opinion that the study's contributions to the field of schizophrenia diagnosis will be recognized as notably substantial.
1- When reviewing the relevant literature, I noticed that the diagnostic accuracy of most existing studies is above 80%, which is already considered an excellent model. Therefore, the authors should highlight the innovation of this paper and emphasize the necessity of conducting this study by addressing the shortcomings of current research.
2- Please cite references from the past five years as some of the references are relatively old.
3- Please cite the following reference in line 53 of the article.

Jauhar S, Johnstone M, McKenna PJ. Schizophrenia. Lancet. 2022 Jan 29;399(10323):473-486. doi: 10.1016/S0140-6736(21)01730-X. PMID: 35093231. Huang Y, Wang Y, Wang H, et al. Prevalence of mental disorders in China: a cross-sectional epidemiological study [published correction appears in Lancet Psychiatry. 2019 Apr;6(4):e11]. Lancet Psychiatry. 2019;6(3):211-224. doi:10.1016/S2215-0366(18)30511-X
4- This study encompasses individuals diagnosed with schizophrenia. It is crucial to comprehensively detail the medical histories, medication regimens, and symptom severities of these patients. Furthermore, it must be determined whether such confounding factors ought to be integrated into the model for the sake of accuracy and control. Additionally, the significance of these metrics in psychiatric research cannot be overstated, raising the question of whether the model adequately adjusts for critical demographic variables such as age and gender.
5- The sample size in this study is relatively small. Should sample size estimation be conducted? Also, considering the inclusion of numerous feature variables, would this potentially lead to poor generalization ability of the model?
6- The discussion section can extract which characteristics of the brain are reflected by these 5 features, and how the model can be further applied in clinical auxiliary diagnosis.

---

## Round 0.2 · Major Revisions

Dear Authors,

Your paper has been re-reviewed. Based on the reviewers' opinions, it must be improved before being considered for publication in PeerJ Computer Science.
In particular, one of the reviewers argued that the study's small sample size impedes its clinical applicability. Accordingly, simple differentiation from a healthy control group does not suffice for clinical diagnostic support. Thus, it would be prudent to refine this article's research objectives, as achieving them in their current form appears unlikely.

This Editor advises the authors to face the above problem in the revised version of their paper.

Reviewer 1 ·

Basic reporting

no comments

Experimental design

no comments.

Validity of the findings

no comments.

Additional comments

Schizophrenia diagnosis based on diverse epoch size using EEG signal on resting-state: Machine learning techniques.
Pls consider the following alternative title:
Schizophrenia diagnosis based on diverse epoch size resting-state EEG using machine learning

Reviewer 2 ·

Basic reporting

See below.

Experimental design

See below.

Validity of the findings

See below.

Additional comments

Many of the issues noted in the author's last revision have not been substantially addressed, and numerous problems persist in the manuscript, necessitating further modifications.
1-The author initially set seven overly ambitious research objectives in the introduction of the article, which were not ultimately realized. Challenges such as the complexity of EEG data collection and the sophisticated processing required by machine learning rendered the goal of using the study as a diagnostic aid for psychiatric conditions or specifically for schizophrenia unfeasible. Additionally, the small sample size of the study impedes its clinical applicability. As a psychiatrist, I contend that for such tools to be clinically viable, they must be validated through substantial sample sizes and tested across various stages of schizophrenia and within diverse psychiatric populations. Simple differentiation from a healthy control group does not suffice for clinical diagnostic support. Thus, it would be prudent to refine the research objectives of this article, as achieving them in their current form appears unlikely.
2- The reference format on line 56 is different from the others. The reference on line 872, "Jauhar, S., M. Johnstone and P. J. McKenna 'FIDMAG informa,'" was not provided by me; please verify it. Additionally, the two references I provided were not fully cited.
3- The order of the tables is somewhat chaotic; the table below Table 2 has a header labeled as Table 1. Additionally, the unit "year" should be placed in the "feature" column.
4- Add the full name of PSDL on line 731.
5- The author's response to the last inquiry stated, "please see the discussion (line 839 to 845)," yet I could not identify the corresponding modifications in the original manuscript. It would enhance the paper if the discussion section were to delve into these specific features, especially discussing how they reflect particular brain functions like learning and memory. Furthermore, incorporating discussions related to clinical aspects would significantly enrich the article's appeal.
6- From what I understand, schizophrenia is associated with frontal lobe dysfunction. It is unclear to me how the author chose the specific five and eight electrodes. Analyzing data from electrodes aligned with particular brain regions might prove more insightful, particularly given that the selected electrodes are not situated in crucial areas of the brain. Even if the findings are favorable, they may not adequately represent the complexities encountered in clinical practice.

---

## Round 0.3 · accepted · Accept

Dear Authors,

Your paper has been revised. Based on the reviewer's report, the paper has been accepted for publication in PEERJ Computer Science. I recommend that you fix all the typos in your manuscript before you submit it in its final form.

Reviewer 2 ·

Basic reporting

No.

Experimental design

No.

Validity of the findings

No.

Additional comments

The comments raised last time have been basically modified, and the following are some remaining problems in the format:

1. On line 193, add the full name of AWGN. Ensure that all abbreviations appearing for the first time in the article are accompanied by their full names.

2. The order of the tables is somewhat confusing.